# Differentiable Quadratic Optimization
# For The Maximum Independent Set Problem

**Ismail R. Alkhouri** [1 2]   **Cedric Le Denmat** [3]
**Yingjie Li** [4]   **Cunxi Yu** [4]   **Jia Liu** [3]   **Rongrong Wang** [2]   **Alvaro Velasquez** [5]

## Abstract

Combinatorial Optimization (CO) addresses many important problems, including the challenging Maximum Independent Set (MIS) problem. Alongside exact and heuristic solvers, differentiable approaches have emerged, often using continuous relaxations of quadratic objectives. Noting that an MIS in a graph is a Maximum Clique (MC) in its complement, we propose a new quadratic formulation for MIS by incorporating an MC term, improving convergence and exploration. We show that every maximal independent set corresponds to a local minimizer, derive conditions with respect to the MIS size, and characterize stationary points. To tackle the non-convexity of the objective, we propose optimizing several initializations in parallel using momentum-based gradient descent, complemented by an efficient MIS checking criterion derived from our theory. We dub our method as **p**arallelized **C**lique-Informed **Q**uadratic **O**ptimization for MIS (pCQO-MIS). Our experimental results demonstrate the effectiveness of the proposed method compared to exact, heuristic, sampling, and data-centric approaches. Notably, our method avoids the out-of-distribution tuning and reliance on (un)labeled data required by data-centric methods, while achieving superior MIS sizes and competitive runtime relative to their inference time. Additionally, a key advantage of pCQO-MIS is that, unlike exact and heuristic solvers, the run-time scales only with the number of nodes in the graph, not the number of edges. Our code is available at the GitHub repository (pCQO-MIS).

[1]University of Michigan, Ann Arbor [2]Michigan State University [3]Ohio State University [4]University of Maryland, College Park [5]University of Colorado, Boulder. Correspondence to: Ismail R. Alkhouri <ismailal@umich.edu; alkhour3@msu.edu>.

*Proceedings of the $42^{nd}$ International Conference on Machine Learning*, Vancouver, Canada. PMLR 267, 2025. Copyright 2025 by the author(s).

## 1. Introduction

In his landmark paper (Karp, 1972), Richard Karp established a connection between Combinatorial Optimization Problems (COPs) and the NP-hard complexity class, implying their inherent computational challenges. Additionally, Richard Karp introduced the concept of reducibility among combinatorial problems that are complete for the complexity class NP.

Although there exists a direct reduction between some COPs – such as the case with the Maximum Independent Set (MIS), Maximum Clique (MC), and Minimum Vertex Cover (MVC) – which allows a solution for one problem to be directly used to solve another, other COPs differ significantly. For example, there exists no straightforward reduction between MIS and the Kidney Exchange Problem (KEP) (McElfresh et al., 2019) (or the Traveling Salesman Problem (TSP) (Dantzig et al., 1954)).

In this paper, we focus on the MIS problem, one of the most fundamental in combinatorial optimization, with many applications including frequency assignment in wireless networks (Matsui & Tokoro, 2000), task scheduling (Eddy & Kochenderfer, 2021), and genome sequencing (Joseph et al., 1992; Zweig et al., 2006).

The MIS problem involves finding a subset of vertices in a graph $G = (V, E)$ with maximum cardinality, such that no two vertices in this subset are connected by an edge (Tarjan & Trojanowski, 1977). In the past few decades, in addition to commercial Integer Programming (IP) solvers (e.g., CPLEX (IBM), Gurobi (Gurobi), and most recently CP-SAT (Perron & Didier)), powerful heuristic methods (e.g., ReduMIS in (Lamm et al., 2016)) have been introduced to tackle the complexities inherent in the MIS problem. Other solvers can be broadly classified into branch-and-bound-based global optimization methods (Akiba & Iwata, 2016), and approximation algorithms (Boppana & Halldórsson, 1992).

More recently, differentiable approaches have been explored (Bengio et al., 2021), falling into two main categories: (*i*) data-driven methods, where a neural network (NN) is trained to fit a distribution over training graphs, and (*ii*) dataless

methods (Alkhouri et al., 2022; Schuetz et al., 2022). Both approaches rely on some formulations of the MIS problem, such as the continuous relaxation of the MIS Quadratic Unconstrained Binary Optimization (QUBO) or ReLU-based objective functions. However, data-driven methods often suffer from unsatisfactory *generalization* performance when faced with graph instances whose structural characteristics differ from those in the training dataset (Böther et al., 2022; Gamarnik, 2023).

In this paper, we present a new differentiable dataless solver for the MIS problem based on an improved quadratic optimization formulation, a parallel optimization strategy, and momentum-based gradient descent, which we dub as **p**arallelized **C**lique-Informed **Q**uadratic **O**ptimization for the **MIS** problem (pCQO-MIS). The contributions of our work are summarized as follows:

1. **MIS Quadratic Formulation with MC Term**: Leveraging the direct relationship between the MIS and MC problems, we propose a new formulation that incorporates an MC term into the continuous relaxation of the MIS quadratic formulation.

2. **Theoretically**:
   - We derive a sufficient and necessary condition for the parameter that penalizes the inclusion of adjacent nodes and the MC term parameter with respect to (w.r.t.) the MIS size.
   - We show that all local minimizers are binary vectors that sit on the boundary of the box constraints, and establish that all these local minimizers correspond to maximal independent sets.
   - We prove that if non-binary stationary points exist, they are saddle points and not local minimizers, with their existence depending on the graph type and connectivity.

3. **Optimization Strategy**: To improve exploration with our non-convex optimization, we propose the use of GPU parallel processing of several initializations for each graph instance using projected momentum-based gradient descent.

4. **Efficient MIS Checking**: Drawing from our theoretical results on local minimizers and stationary points, we develop an efficient MIS checking function that significantly accelerates our implementation.

5. **Experimental Validation**: We evaluate our approach on challenging benchmark graph datasets, demonstrating its efficacy. Our method achieves competitive or superior performance compared to state-of-the-art heuristic, exact, and data-driven approaches in terms of MIS size and/or run-time.

## 2. Preliminaries

**Notations:** Consider an undirected graph represented as $G = (V, E)$, where $V$ is the vertex set and $E \subseteq V \times V$ is the edge set. The number of nodes (resp. edges) is denoted by $|V| = n$ (resp. $|E| = m$), where $| \cdot |$ denotes the cardinality of a set. Unless otherwise stated, for a node $v \in V$, we use $\mathcal{N}(v) = \{u \in V \mid (u,v) \in E\}$ to denote the set of its neighbors. The degree of a node $v \in V$ is denoted by $\mathrm{d}(v) = |\mathcal{N}(v)|$, and the maximum degree of the graph by $\Delta(G)$. For a subset of nodes $U \subseteq V$, we use $G[U] = (U, E[U])$ to represent the subgraph induced by the nodes in $U$, where $E[U] = \{(u,v) \in E \mid u,v \in U\}$. Given a graph $G$, its complement is denoted by $G' = (V, E')$, where $E' = V \times V \setminus E$ is the set of all the edges between nodes that are not connected in $G$. Consequently, if $|E'| = m'$, then $m + m' = n(n-1)/2$ represents the number of edges in the complete graph on $V$. For any $v \in V$, $\mathcal{N}'(v) = \{u \in V \mid (u,v) \in E'\}$ denotes the neighbor set of $v$ in the complement graph $G' = (V, E')$. The adjacency matrix of graph $G$ is denoted by $\mathbf{A}_G \in \{0,1\}^{n \times n}$. We use $\mathbf{I}$ to denote the identity matrix. The trace of a matrix $\mathbf{A}$ is denoted by $\mathrm{tr}(\mathbf{A})$. For any positive integer $n$, $[n] := \{1, \dots, n\}$. The vector (resp. matrix) of all ones and size $n$ (resp. $n \times n$) is denoted by $\mathbf{e}_n$ (resp. $\mathbf{J}_n = \mathbf{e}_n \mathbf{e}_n^T$). Furthermore, we use $\mathbb{1}(\cdot)$ to denote the indicator function that returns 1 (resp. 0) when its argument is True (resp. False).

**Problem Statement:** In this paper, we consider the NP-hard problem of obtaining the maximum independent set (MIS). Next, we formally define MIS and the complementary Maximum Clique (MC) problems.

**Definition 1** (MIS Problem). Given an undirected graph $G = (V, E)$, the goal of MIS is to find a subset of vertices $\mathcal{I} \subseteq V$ such that $E([\mathcal{I}]) = \emptyset$, and $|\mathcal{I}|$ is maximized.

**Definition 2** (MC Problem). Given an undirected graph $G = (V, E)$, the goal of MC is to find a subset of vertices $\mathcal{C} \subseteq V$ such that $G[\mathcal{C}]$ is a complete graph, and $|\mathcal{C}|$ is maximized.

For the MC problem, the MIS of a graph is an MC of the complement graph (Karp, 1972). This means that MIS $\mathcal{I}$ in $G$ is equivalent to MC $\mathcal{C}$ in $G'$.

Given a graph $G$, if $\mathcal{I}$ is a *Maximal Independent Set (MaxIS)*, then $E([\mathcal{I}]) = \emptyset$, but $|\mathcal{I}|$ is not necessarily the largest in $G$. If $\mathcal{I}$ is an *Independent Set (IS)* that it is not maximal, then $E([\mathcal{I}])$ is an empty set, but there exists at least one $v \notin \mathcal{I}$ such that

$$E([\mathcal{I} \cup \{v\}]) = \emptyset \,. \tag{1}$$

See Figure 1 for an example. We note that, in this paper, we use MIS and MaxIS interchangeably.

Let $\mathbf{z}_v$ be an entry of the binary vector $\mathbf{z} \in \{0,1\}^n$ that corresponds to a node $v \in V$. The integer linear program

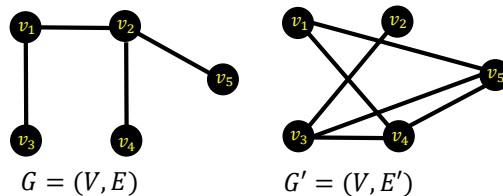

$$G = (V, E) \qquad G' = (V, E')$$

Figure 1: A graph $G$ (*left*) and its complement graph $G'$ (*right*). Sets $\text{MIS}_1 = \{v_1, v_4, v_5\}$ and $\text{MIS}_2 = \{v_3, v_4, v_5\}$ correspond to a maximum independent set in $G$ and an MC in $G'$. Set $\text{MaxIS} = \{v_2, v_3\}$ corresponds to a maximal independent set as its not of maximum cardinality. Set $\text{IS} = \{v_1, v_4\}$ is not a maximal independent set because $\text{IS} \cup \{v_5\}$ is equivalent to $\text{MIS}_1 = \{v_1, v_4, v_5\}$.

(ILP) for MIS is (Nemhauser & Trotter, 1975):

$$\max_{\mathbf{z} \in \{0,1\}^n} \sum_{v \in V} \mathbf{z}_v \quad \text{s.t.} \quad \mathbf{z}_v + \mathbf{z}_u \leq 1 \,, \forall (v, u) \in E. \quad (2)$$

Furthermore, the following QUBO in (3) (with an optimal solution that is equivalent to the optimal solution of the above ILP) can also be used to formulate the MIS problem (Pardalos & Rodgers, 1992):

$$\max_{\mathbf{z} \in \{0,1\}^n} \mathbf{e}_n^T \mathbf{z} - \frac{\gamma_Q}{2} \mathbf{z}^T \mathbf{A}_G \mathbf{z} \,, \quad (3)$$

where $\gamma_Q > 0$ is a parameter that penalizes the selection of two nodes with an edge connecting them. In (Mahdavi Pajouh et al., 2013), it was shown that the condition $\gamma_Q > 1$ is both sufficient and necessary for local minimizers to correspond to binary vectors representing MaxISs.

In Appendix D, we review various approaches for solving the MIS problem.

## 3. Clique-Informed Differentiable Quadratic MIS Optimization

In this section, we first introduce the clique-informed quadratic optimization (CQO) formulation for the MIS problem. Next, we provide theoretical insights into the objective function, and then present our parallelized optimization strategy using momentum-based gradient descent (MGD).

### 3.1. Optimization Reformulation

Our proposed optimization reformulation is

$$\min_{\mathbf{x} \in [0,1]^n} f(\mathbf{x}) := -\mathbf{e}_n^T \mathbf{x} + \frac{\gamma}{2} \mathbf{x}^T \mathbf{A}_G \mathbf{x} - \frac{\gamma'}{2} \mathbf{x}^T \mathbf{A}_{G'} \mathbf{x} \,, \quad (4)$$

where $\gamma > 1$, analogous to $\gamma_Q$ in (3), serves as the edge penalty parameter. The third term represents the maximum clique (MC) term we propose in this paper, with parameter

$\gamma' \geq 1$, introduced to discourage sparsity in the solution. The function $f(\mathbf{x})$ can also be expressed as

$$f(\mathbf{x}) = -\sum_{v \in V} \mathbf{x}_v + \gamma \sum_{(u,v) \in E} \mathbf{x}_v \mathbf{x}_u - \gamma' \sum_{(u,v) \in E'} \mathbf{x}_v \mathbf{x}_u \,.$$

Utilizing the identity

$$\mathbf{A}_{G'} = \mathbf{J}_n - \mathbf{I} - \mathbf{A}_G \,, \quad (5)$$

$\mathbf{x}^T \mathbf{x} = \|\mathbf{x}\|_2^2$, and the non-negative entries of $\mathbf{x}$ for which we can write $\|\mathbf{x}\|_1^2 = \mathbf{x}^T \mathbf{J}_n \mathbf{x}$, our proposed function can be rewritten as

$$f(\mathbf{x}) = -\mathbf{e}_n^T \mathbf{x} + \frac{\gamma + \gamma'}{2} \mathbf{x}^T \mathbf{A}_G \mathbf{x} + \frac{\gamma'}{2}(\|\mathbf{x}\|_2^2 - \|\mathbf{x}\|_1^2). \quad (6)$$

In particular, we incorporate $\gamma$ that penalizes edges in graph $G$ on the optimization objective. The third term is informed by the duality between the MIS and MC problems.

The rationale behind the third term $-\frac{\gamma'}{2} \mathbf{x}^T \mathbf{A}_{G'} \mathbf{x}$ in (4) (corresponding to the edges of the complement graph $G'$) is to (*i*) encourage the optimizer to select two nodes with no edge connecting them in $G$ (implying an edge in $G'$), and (*ii*) discourage sparsity given the last term of (6).

Let $\mathbf{z}^*$ be a binary minimizer of (4) with

$$\mathcal{I}(\mathbf{z}^*) = \{v \in V : \mathbf{z}_v^* = 1\} \quad (7)$$

Then, we have:

$$f(\mathbf{z}^*) = -\sum_{v \in V} \mathbb{1}(\mathbf{z}_v^* = 1) - \gamma' |E'([\mathcal{I}(\mathbf{z}^*)])| \,. \quad (8)$$

This expression includes only the first and third terms, as there are no edges connecting any two nodes in $\mathcal{I}(\mathbf{z}^*)$.

**Remark 1.** Given that the number of non-zero entries in $\mathbf{A}_G$ is $2m$ (with one entry for each edge in $G$ and $\mathbf{A}_G$ being symmetric), the computational cost of a continuous relaxation of the QUBO formulation in (3) (with box constraints) is $\mathcal{O}(mn)$. Because the vector-matrix multiplication in (6) is only in the second term, the computational cost of our proposed function is also $\mathcal{O}(mn)$. This means that including the MC term in our proposed objective results in the same computational cost as (3) with $[0, 1]^n$.

### 3.2. Theoretical Insights

In this subsection, we provide theoretical insights where we first examine the constant Hessian of $f(\mathbf{x})$ in (4). Then, we provide the necessary and sufficient condition for $\gamma$ and $\gamma'$ for any MaxIS to correspond to local minimizers of (4). Moreover, we also provide a sufficient condition for all local minimizers of (4) to be associated with a MaxIS. Additionally, we show that if non-binary stationary points exist, they are saddle points. We relegate the detailed proofs to Appendix A.

**Definition 3** (MaxIS vector). Given a graph $G = (V, E)$, a binary vector $\mathbf{x} \in \{0,1\}^n$ is called a MaxIS vector if there exists a MaxIS $\mathcal{I}$ of $G$ such that $\mathbf{x}_i = 1$ for all $i \in \mathcal{I}$, and $\mathbf{x}_i = 0$ for all $i \notin \mathcal{I}$.

**Lemma 1.** *For any non-complete graph $G$, the constant hessian of $f(\mathbf{x})$ in* (4)*, i.e., $\gamma \mathbf{A}_G - \gamma' \mathbf{A}_{G'}$, is always a non-positive-semidefinite (non-PSD) matrix.*

*Proof Sketch:* Here, we show that the Hessian is a non-PSD matrix by showing that for any MaxIS vector $\mathbf{x}$, the condition $\mathbf{x}^T(\gamma \mathbf{A}_G - \gamma' \mathbf{A}_{G'})\mathbf{x} \geq 0$ can not be satisfied.

The result in Lemma 1 indicates that our quadratic optimization problem is always non-convex for any non-complete graph. The work in (Burer & Letchford, 2009) discusses the complexity of box-constrained continuous non-convex quadratic optimization problems.

**Theorem 1** (Necessary and Sufficient Condition on $\gamma$ and $\gamma'$ for MaxIS vectors to be local minimizers of (4)). *Given an arbitrary graph $G = (V, E)$ and its corresponding formulation in* (4)*, suppose the size of any MIS of $G$ is $k$. Then, $\gamma \geq 1 + \gamma' k$ is necessary and sufficient for all MaxIS vectors to be local minimizers of* (4) *for arbitrary graphs.*

*Proof Sketch:* Given a MaxIS $\mathcal{I}$ with $|\mathcal{I}| = k$, we derive the bound by considering the boundary points enforced by the box-constraints, and the gradient of $f(\mathbf{x})$ w.r.t. some $v \in V \setminus \mathcal{I}$.

**Remark 2.** Theorem 1 offers guidance on selecting $\gamma$ and $\gamma'$. While the MIS set size $k$ is typically unknown in advance, it's possible to use classical estimates of $k$ to inform the choice of these parameters. For example, as shown in (Wei, 1981), $k$ can be bounded by

$$k \geq \sum_{v \in V} \frac{1}{1 + \mathrm{d}(v)} \,, \tag{9}$$

which could provide a useful estimate for this purpose.

Next, we provide further characterizations of the local minimizers of (4).

**Lemma 2.** *All local minimizers of* (4) *are binary vectors.*

*Proof Sketch:* We prove this by showing that for any coordinates of $\mathbf{x}$ with non-binary values, one necessary condition for any local minimizer can not be satisfied.

Building on the result of Lemma 2, we provide a stronger condition on $\gamma$ and $\gamma'$ that ensures all local minimizers of (5) correspond to a MaxIS.

**Theorem 2** (Local Minimizers of (4)). *Given graph $G = (V, E)$ and set $\gamma > 1 + \gamma' \Delta(G')$, all local minimizers of* (4) *are MaxIS vectors of $G$.*

*Proof Sketch:* By Lemma 2, we examine the local minimizers that are binary. With this, we prove that all local minimizers are ISs. Then, we show that any IS, that is not maximal, is a not a local minimizer.

**Remark 3.** The assumption $\gamma > 1 + \gamma' \Delta(G')$ in Theorem 2 is stronger than that in Theorem 1. The trade-off of selecting a larger $\gamma$ value is that, while it ensures that only MaxISs are local minimizers, it also increases the non-convexity of the optimization problem, making it more challenging to solve.

**Remark 4.** Although the proposed box-constrained quadratic Problem (4) is still NP-hard to solve for the global minimizer(s), it is a relaxation of the original integer programming problem. It can leverage gradient information, allowing the use of high-performance computational resources and parallel processing to enhance the efficiency and scalability of our approach.

In the following theorem, we provide results regarding points where the gradient of $f(\mathbf{x})$ is zero.

**Theorem 3** (Non-Extremal Stationary Points). *For any graph $G$, assume that there exists a point $\mathbf{x}'$ such that $\nabla_{\mathbf{x}} f(\mathbf{x}') = \mathbf{0}$, i.e.,*

$$\mathbf{x}' := (\gamma \mathbf{A}_G - \gamma' \mathbf{A}_{G'})^{-1} \mathbf{e}_n \,. \tag{10}$$

*Then, $\mathbf{x}'$ is not a local minimizer of* (4) *and therefore does not correspond to a MaxIS.*

*Proof Sketch:* We show that $\mathbf{x}'$ is not a local minimizer by showing that it can not be binary, building upon the result on Lemma 2.

**Remark 5.** The above theorem implies that while there may exist a non-binary stationary point $\mathbf{x}'$, it is a *saddle point*, not a local minimizer, as indicated by the zero gradient vector and by Lemma 1 (the Hessian is always non-PSD). Momentum-based Gradient Descent (MGD) is typically effective at escaping saddle points and converging to local minimizers, which serves as one motivation of its use in pCQO-MIS. Furthermore, we observe that this specific saddle point is never encountered in our empirical evaluations and that it depends on the structure of the graph. In many instances, it lies outside the box constraints, depending on the graph's density and connectivity. Further discussion about the existence of saddle points is provided in Appendix C.

### 3.3. Optimization Strategy

Given the highly non-convex nature of our optimization problem, this section introduces the pCQO-MIS method for efficiently obtaining MaxISs. We first describe the projected MGD and parallel initializations used. Then, we present the efficient MaxIS checking criterion, followed by a detailed outline of the algorithm.

### 3.3.1. PROJECTED MOMENTUM-BASED GRADIENT DESCENT

As previously discussed, our function in (4) is highly non-convex which makes finding the global minimizer(s) a challenging task. However, first-order gradient-based optimizers are effective for finding a local minimizer given an initialization in $[0,1]^n$.

Given the full differentiability of the objective in (4), with the gradient vector defined as

$$\mathbf{g}(\mathbf{x}) := \nabla_{\mathbf{x}} f(\mathbf{x}) = -\mathbf{e}_n + (\gamma \mathbf{A}_G - \gamma' \mathbf{A}_{G'})\mathbf{x}, \quad (11)$$

MGD empirically proves to be computationally efficient. Specifically, let $\mathbf{v} \in \mathbb{R}^n$, $\beta \in (0,1)$, and $\alpha > 0$ represent the velocity vector, momentum parameter, and optimization step size for MGD, respectively.

The projected MGD (Polyak, 1964) updates are then defined as follows:

$$\mathbf{v} \leftarrow \beta \mathbf{v} + \alpha \mathbf{g}(\mathbf{x}), \quad (12a)$$

$$\mathbf{x} \leftarrow \text{Proj}_{[0,1]^n}(\mathbf{x} - \mathbf{v}). \quad (12b)$$

We implement the updates in (12) based on our empirical observation that fixed-step-size gradient descent for (4) is sensitive to the choice of step size and frequently fails to converge to local minimizers due to overshooting. This serves as another motivation of why we adopt Momentum-based Gradient Descent (MGD), as further supported in Appendix E.4.

### 3.3.2. DEGREE-BASED PARALLEL INITIALIZATIONS

For a single graph, we propose to use various points in $[0,1]^n$ and execute the updates in (12) in parallel for each. Given a specified number of parallel processes $M$, we define $S_{\text{ini}}$ to denote the set of multiple initializations, where $|S_{\text{ini}}| = M$.

Based on the intuition that vertices with higher degrees are less likely to belong to an MIS compared to those with lower degrees (Alkhouri et al., 2022), we initialize $S_{\text{ini}}$ with $M$ samples drawn from a Gaussian distribution $\mathcal{N}(\mathbf{m}, \eta \mathbf{I})$. Here, $\mathbf{m}$ is the mean vector, initially set to $\mathbf{h}$, where $\mathbf{h}$ is:

$$\mathbf{h}_v = 1 - \frac{\mathrm{d}(v)}{\Delta(G)}, \forall v \in V, \quad (13)$$

$$\mathbf{h} \leftarrow \frac{\mathbf{h}}{\max_v \mathbf{h}_v}.$$

$\eta$ is a hyper-parameter that regulates the exploration around $\mathbf{m}$. Once the optimization for each initialization is complete, we proceed with the MaxIS checking procedure for all the results, which we discuss next.

### 3.3.3. EFFICIENT IMPLEMENTATION OF MAXIMAL IS CHECKING

Given a binary vector $\mathbf{z} \in \{0,1\}^n$ with

$$\mathcal{I}(\mathbf{z}) := \{v \in V : \mathbf{z}_v = 1\}, \quad (14)$$

the standard approach to check whether it is an IS and then whether it is a MaxIS involves iterating over all nodes to examine their neighbors. Specifically, this entails verifying that (*i*) no two nodes $(v, u) \in E$ with $\mathbf{z}_v = \mathbf{z}_u = 1$ exist (IS checking), and (*ii*) there does not exist any $u \notin \mathcal{I}(\mathbf{z})$ such that $\forall w \in \mathcal{I}(\mathbf{z})$, $u \notin \mathcal{N}(w)$ (MaxIS checking). However, as the order and density of the graph increase, the computational time required for this process may become significantly longer.

Matrix-vector multiplication can be used for IS checking, as the condition $\mathbb{1}(\mathbf{z}^T \mathbf{A}_G \mathbf{z} = 0)$ indicates the presence of edges in the graph. If $\mathbf{z}^T \mathbf{A}_G \mathbf{z} > 0$, then $\mathbf{z}$ can be immediately identified as not being an IS. While this approach efficiently checks for IS validity, it cannot determine whether the IS is maximal.

Building on the characteristics of local minimizers and the non-extremal stationary points of (4), discussed in Lemma 2, Theorem 2, and Theorem 3, we propose an efficient implementation for checking whether a vector $\mathbf{x} \in [0,1]^n$ corresponds to a MaxIS.

Specifically, Lemma 2, demonstrates that all local minimizers are binary. Subsequently, in Theorem 2, we establish that all local minimizers correspond to MaxISs. This implies that all binary stationary points resulting from the updates in (12) within our box-constrained optimization in (4) are local minimizers situated at the boundary of $[0,1]^n$ and correspond to MaxISs, as further elaborated in the proof of Theorem 2. Consequently, we propose a new MaxIS checking condition that relies on a single matrix-vector multiplication. For a given $\mathbf{x} \in [0,1]^n$, we first obtain its binary representation as a vector $\mathbf{z}$, where $\mathbf{z}_v = \mathbb{1}(\mathbf{x}_v > 0)$ for all $v \in V$. We then verify whether the following condition is satisfied.

$$\mathbb{1}\Big(\mathbf{z} = \text{Proj}_{[0,1]^n}\big(\mathbf{z} - \alpha \mathbf{g}(\mathbf{z})\big)\Big). \quad (15)$$

Equation (15) represents a simple projected gradient descent step to determine whether $\mathbf{z}$ is at the boundary of the box-constraints. If (15) holds true, then the MaxIS is given by $\mathcal{I}(\mathbf{z})$, as defined in (14).

**Remark 6.** As previously discussed, the work in (Mahdavi Pajouh et al., 2013) showed that any binary minimizer of a box-constrained continuous relaxation of (3) corresponds to a MaxIS when $\gamma_{\text{Q}} > 1$. This means that verifying whether a binary vector corresponds to a MaxIS using the proposed projected gradient descent step can also be applied

**Algorithm 1** pCQO-MIS.

**Input**: Graph $G = (V, E)$, set of initializations $S_{\text{ini}}$, number of iterations $T$ per one initialization, edge-penalty parameter $\gamma$, MC term parameter $\gamma'$, and MGD parameters: Step size $\alpha$, and momentum parameter $\beta$.
**Output**: The best obtained MaxIS $\mathcal{I}^*$ in $G$

01: **Initialize** $S_{\text{MaxIS}} = \{\cdot\}$ (Empty set to collect MaxISs)
02: **For** $\mathbf{x}[0] \in S_{\text{ini}}$ (**Parallel Execution**)
03:     **Initialize** $\mathbf{v}[0] \leftarrow \mathbf{0}$
04:     **For** $t \in [T]$
05:         **Obtain** $\mathbf{g}(\mathbf{x}[t-1]) = -\mathbf{e}_n + (\gamma \mathbf{A}_G - \gamma' \mathbf{A}_{G'})\mathbf{x}[t-1]$
06:         **Obtain** $\mathbf{v}[t] = \beta \mathbf{v}[t-1] + \alpha \mathbf{g}(\mathbf{x}[t-1])$
07:         **Obtain** $\mathbf{x}[t] = \text{Proj}_{[0,1]^n}(\mathbf{x}[t-1] - \mathbf{v}[t])$
08:     **Obtain** $\mathbf{z}[T]$ with $\mathbf{z}_v[T] = \mathbb{1}(\mathbf{x}_v[T] > 0), \forall v \in V$
09:     **If** $\mathbb{1}\big(\mathbf{z}[T] = \text{Proj}_{[0,1]^n}(\mathbf{z}[T] - \alpha \mathbf{g}(\mathbf{z}[T]))\big)$
10:         **Then** $S_{\text{MaxIS}} \leftarrow S_{\text{MaxIS}} \cup \mathcal{I}(\mathbf{z}[T])$
11: **Return** $\mathcal{I}^* = \text{argmax}_{\mathcal{I} \in S_Q} |\mathcal{I}|$

using (3) as:

$$\mathbb{1}\Big(\mathbf{z} = \text{Proj}_{[0,1]^n}\big(\mathbf{z} + \alpha(\mathbf{e}_n - \gamma_Q \mathbf{A}_G \mathbf{z})\big)\Big). \tag{16}$$

In Section 4.4, we show the speedups obtained from using this approach as compared to the standard iterative approach discussed earlier in this subsection.

### 3.3.4. THE pCQO-MIS ALGORITHM

We outline the proposed procedure in Algorithm 1. As shown, the algorithm takes a graph $G$, the set of initializations $S_{\text{ini}}$, the maximum number of iterations per batch $T$ (with iteration index $t$), the edge penalty parameter $\gamma$, the MC term parameter $\gamma'$, step size $\alpha$, and momentum parameter $\beta$ as inputs.

For each initialization vector in set $S_{\text{ini}}$ and iteration $t \in [T]$, Lines 5 to 7 involve updating the optimization variable $\mathbf{x}[t]$. After $T$ iterations, in Lines 8 to 10, the algorithm checks whether the binary representation of $\mathbf{x}[T]$ corresponds to a MaxIS using (15). Finally, the best-found MaxIS, determined by its cardinality, is returned in Line 10.

After $M > 1$ optimizations are complete (i.e., when the batch is complete), another set of initializations are placed in $S_{\text{ini}}$. Then Algorithm 1 is executed again, depending on the time budget and the availability of the computational resources (number of batches). When Algorithm 1 is executed again, the vector $\mathbf{v}$ is not re-initialized, but rather maintained from the previous batch. Subsequent runs depend on sampling from $\mathcal{N}(\mathbf{m}, \eta\mathbf{I})$ where $\mathbf{m}$ is set to the binarized vector of the best obtained MaxIS from the previous run.

**Remark 7.** Optimizing initialized points around a binary vector that corresponds to a MaxIS shows that pCQO-MIS can be used as a local search heuristic for MIS.

**Remark 8.** While Theorem 2 indicates how to select $\gamma$ and $\gamma'$, other hyper-parameters (i.e., $\alpha, \beta$, and $T$) still need to be tuned to obtain feasible solutions. In Appendix E.9.1, we describe a basic grid search procedure to select these parameters.

## 4. Experimental results

### 4.1. Settings, Baselines, & Benchmarks

We code our objective function and the proposed algorithm using C++. For baselines, we utilize Gurobi (Gurobi) and the recent Google solver CP-SAT (Perron & Didier) for the ILP in (2), ReduMIS (Lamm et al., 2016), iSCO[1] (Sun et al., 2023), and four learning-based methods: DIMES (Qiu et al., 2022), DIFUSCO (Sun & Yang, 2023), LwD (Ahn et al., 2020), and the GCN method in (Li et al., 2018) (commonly referred to as 'Intel'). We note that, following the analysis in (Böther et al., 2022), GCN's code cloning to ReduMIS is disabled, which was also done in (Qiu et al., 2022; Sun & Yang, 2023). To show the impact of the MC term, we include the results of pCQO-MIS without the third term (i.e., $\gamma' = 0$) which we term pQO-MIS (see also Appendix E.5).

Aligned with recent methods (DIMES, DIFUSCO, and iSCO), we employ the Erdos-Renyi (ER) (Erdos et al., 1960) graphs from (Qiu et al., 2022) and the SATLIB graphs from (Hoos & Stützle, 2000) as benchmarks. The ER dataset[2] consists of 128 graphs with 700 to 800 nodes and $p = 0.15$, where $p$ is the probability of edge creation. The SATLIB dataset consists of 500 graphs (with at most $1{,}347$ nodes and $5{,}978$ edges). Additionally, the GNM random graph generator function of NetworkX (Hagberg et al., 2008) is utilized for our scalability experiment in Section 4.3. Results for the DIMACS graphs (Johnson & Trick, 1996), larger ER graphs, and the BA graphs from (Wu et al., 2025) are given in Appendix E.1, Appendix E.2, and Appendix E.3, respectively.

For pCQO-MIS, the hyper-parameters are set as given in Table 12 of Appendix E.9. Further implementation details and results are provided in Appendix E. Our code is available online[3].

### 4.2. ER and SATLIB Benchmark Results

Here, we present the results of pCQO-MIS alongside the considered baselines, using the SATLIB and ER benchmarks. These results are measured in terms of average MIS size across the graphs in the dataset and the total sequential run-time (in minutes) required to obtain the results for all the graphs. Results are given in Table 1, where the last 4

---

[1] https://github.com/google-research/discs
[2] https://github.com/DIMESTeam/DIMES
[3] https://github.com/ledenmat/pCQO-mis-benchmark

| Method | Type | Dataset: SATLIB | | | Dataset: ER | | |
|--------|------|-----------------|---|---|-------------|---|---|
| | | Training Data | MIS Size ($\uparrow$) | Run-time ($\downarrow$) | Training Data | MIS Size ($\uparrow$) | Run-time ($\downarrow$) |
| ReduMIS (Lamm et al., 2016) | Heuristics | $\times$ | **425.96** | 37.58 | $\times$ | 44.87 | 52.13 |
| CP-SAT (Perron & Didier) | Exact | $\times$ | **425.96** | 0.56 | $\times$ | 41.15 | 64 |
| Gurobi (Gurobi) | Exact | $\times$ | **425.96** | 8.32 | $\times$ | 39.14 | 64 |
| GCN (Li et al., 2018) | SL+G | SATLIB | 420.66 | 23.05 | SATLIB | 34.86 | 23.05 |
| LwD (Ahn et al., 2020) | RL+S | SATLIB | 422.22 | 18.83 | ER | 41.14 | 6.33 |
| DIMES (Qiu et al., 2022) | RL+TS | SATLIB | 423.28 | 20.26 | ER | 42.06 | 12.01 |
| DIFUSCO (Sun & Yang, 2023) | RL+G | SATLIB | 424.5 | 8.76 | ER | 38.83 | 8.8 |
| DIFUSCO (Sun & Yang, 2023) | RL+S | SATLIB | 425.13 | 23.74 | ER | 41.12 | 26.27 |
| iSCO (Sun et al., 2023) | S | $\times$ | 422.664 | "22.35" | $\times$ | 44.57 | "14.88" |
| pQO-MIS (i.e., $\gamma' = 0$) | QO | $\times$ | 412.888 | 16.964 | $\times$ | 40.398 | 5.78 |
| **pCQO-MIS** | QO | $\times$ | 425.148 | 56.722 | $\times$ | **45.109** | 54.766 |
| **pCQO-MIS** | QO | $\times$ | 424.686 | 31.901 | $\times$ | 45.078 | 40.555 |
| **pCQO-MIS** | QO | $\times$ | 424.096 | 20.3 | $\times$ | 44.969 | 20.875 |
| **pCQO-MIS** | QO | $\times$ | 423.706 | 16.394 | $\times$ | 44.5 | 5.563 |

Table 1: Benchmark dataset results in terms of **average MIS size** and **total sequential run-time** (minutes). RL, SL, G, QO, S, and TS represent Reinforcement Learning, Supervised Learning, Greedy decoding, Quadratic Optimization, Sampling, and Tree Search, respectively. The results of the learning-based methods (other than DIFUSCO) and ReduMIS are sourced from (Qiu et al., 2022) and run using a single NVIDIA A100 40GB GPU and AMD EPYC 7713 CPU. The results of DIFUSCO are sourced from (Sun & Yang, 2023) and run using a single NVIDIA V100 GPU and Intel Xeon Gold 6248 CPU. The run-time for learning methods exclude the training time (underlined). pCQO-MIS run-times exclude the hyper-parameters tuning time that was done on one graph for each dataset (see Appendix E.9.1). The pCQO-MIS, CP-SAT, and Gurobi results are run using an NVIDIA RTX3070 GPU and Intel I9-12900K CPU. The results for iSCO were produced using an NVIDIA A100 40GB GPU and AMD EPYC 7H12 CPU. We note that the run time reported in iSCO (Table 1 in (Sun et al., 2023)) is for running multiple graphs in parallel, not a sequential total run time. We evaluated iSCO in the same way. If they are run sequentially, the extrapolated run-time is ~9000 minutes for SATLIB and ~140 minutes for ER. ReduMIS employs the local search procedure from (Andrade et al., 2012) for multiple rounds, which no other method in the table uses, following the study in (Böther et al., 2022). Different run-times for pCQO-MIS correspond to using different number of batches (See Appendix E.10). For more details about the requirements of each method, see Appendix D.1.

rows show the pCQO-MIS results for different run-times. We note that the ER results from the exact solvers are limited to 30 seconds per graph to ensure total run-times that are comparable to those of other methods. In what follows, we provide observations on these results.

- All learning-based methods, except for GCN, require training a separate network for each graph dataset, as shown in the third and sixth columns of Table 1, highlighting their generalization limitations. In contrast, our method is more generalizable, requiring only the tuning of hyper-parameters for each set of graphs. See Appendix E.8 for a comparison between pCQO-MIS and DIFUSCO using graphs with densities that are different from the training setting of DIFUSCO.

- When compared to learning-based approaches, our method outperforms all baseline methods in terms of MIS size, all without requiring any training data. We note that the reported run times for learning-based methods exclude training time, which can vary depending on several factors, including graphs sizes, available computing resources, the number of data points, and the specific neural network architecture used. In under 6 minutes (which

is shorter than the inference time of any learning-based method), pCQO-MIS reports larger MIS sizes than any learning method (44.5 vs. 42.06). Furthermore, our approach does not rely on additional techniques such as Greedy Decoding (Graikos et al., 2022) and Monte Carlo Tree Search (Fu et al., 2021).

- When compared to the sampling approach, iSCO, our method reports larger MIS sizes while requiring significantly reduced sequential run-time. We note that the iSCO paper (Sun et al., 2023) reports a lower run time as compared to other methods. This reported run time is achieved by evaluating the test graphs in parallel, in contrast to all other methods that evaluated them sequentially. To fairly compare methods in our experiments, we opted to report sequential test run time only. We conjecture that the extended sequential run-time of iSCO, compared to its parallel run-time, is due to its use of simulated annealing. Because simulated annealing depends on knowing the energy of the previous step when determining the next step, it is inherently more efficient for iSCO to solve many graphs in parallel than in series.

- For SATLIB, which consists of sparser graphs (relative to

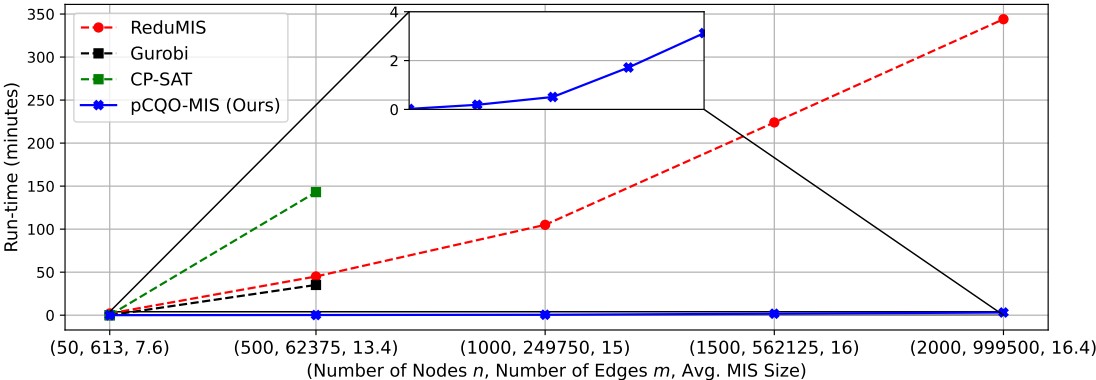

Figure 2: Total run-time in minutes (y-axis) of pCQO-MIS, ReduMIS, CP-SAT, and Gurobi for the **GNM** graphs with $n \in \{50, 500, 1000, 1500, 2000\}$, $m = \lceil \frac{n(n-1)}{4} \rceil$, and the average MIS size of 5 graphs (x-axis). This choice of the number of edges indicates that half of the total possible edges (w.r.t. the complete graph) exist. Here, we also use an NVIDIA RTX3070 GPU and Intel I9-12900K CPU. For $n > 500$, Gurobi and CP-SAT are not included due to excessive run-times.

the ER graphs with $p = 0.15$), pCQO-MIS falls just short when compared to ReduMIS, Gurobi, and CP-SAT (exact and heuristic solvers). The reason ReduMIS achieves SOTA results here is that it applies a large set of MIS-specific graph reductions, along with the 2-opt local search procedure (Andrade et al., 2012). pCQO-MIS and other baselines do not apply the 2-opt procedure following the study in (Böther et al., 2022) where it was conjectured that most methods will converge to the same solutions if this local search procedure is applied (for each solution found). We note that ReduMIS iteratively applies this heuristic. For denser graphs, most of these graph reductions are not applicable. Gurobi and CP-SAT solve the ILP in (2) where the number of constraints is equal to the number of edges in the graph. This means that Gurobi and CP-SAT are expected to perform better on SATLIB, where there are fewer constraints, compared to denser graphs like ER.

- On ER, our method not only reports a larger average MIS size but also generally requires less run-time. Specifically, in under 21 minutes, our method (pCQO-MIS) achieves better results than ReduMIS, CP-SAT, and Gurobi. In under 55 minutes, we achieve **45.109**. We emphasize that we outperform the SOTA MIS heuristic solver and two commercial solvers[4].

- Given the same run-time, when comparing the results of pQO-MIS (i.e., $\gamma' = 0$) and the results of pCQQ-MIS, we observe that when the MC term is included, pCQO-MIS reports larger MIS sizes. On average, using the MC term yields nearly 11 (resp. 4) nodes improvement for SATLIB (resp. ER). A detailed study about the impact of the clique

term is given in Appendix E.5.

### 4.3. Scalability Results

It is well-established that relatively denser graphs pose greater computational challenges compared to sparse graphs. This is due to the applicability of graph reduction techniques such as the LP reduction method in (Nemhauser & Trotter, 1975), and the unconfined vertices rule (Xiao & Nagamochi, 2013) (see (Lamm et al., 2016) for a complete list of the graph reduction rules that apply only on sparse graphs). For instance, by simply applying the LP graph reduction technique, the large-scale highly sparse graphs (with several hundred thousand nodes), considered in Table 5 of (Li et al., 2018), reduce to graphs of a few thousands nodes with often dis-connected sub-graphs that can be treated independently.

Therefore, the scalability and performance of ReduMIS are significantly dependent on the sparsity of the graph. This dependence emerges from the iterative application of various graph reduction techniques (and the 2-opt local search in (Andrade et al., 2012)) in ReduMIS, specifically tailored for sparse graphs. For instance, the ReduMIS results presented in Table 2 of (Ahn et al., 2020) are exclusively based on very large and highly sparse graphs. This conclusion is substantiated by both the sizes of the considered graphs and the corresponding sizes of the obtained MIS solutions. As such, in this subsection, we investigate the scalability of pCQO-MIS against the SOTA methods: ReduMIS, Gurobi, and CP-SAT on denser graphs.

To generate suitably dense graphs, we utilized the NetworkX GNM graph generator with the number of edges set to $m = \lceil \frac{n(n-1)}{4} \rceil$. It is important to note that the density of these graphs is significantly higher than those considered in the previous subsection (and most of the graphs consid-

---

[4]We note that learning-based methods, such as (Qiu et al., 2022; Sun & Yang, 2023), use ReduMIS to label training graphs under the supervised learning setting.

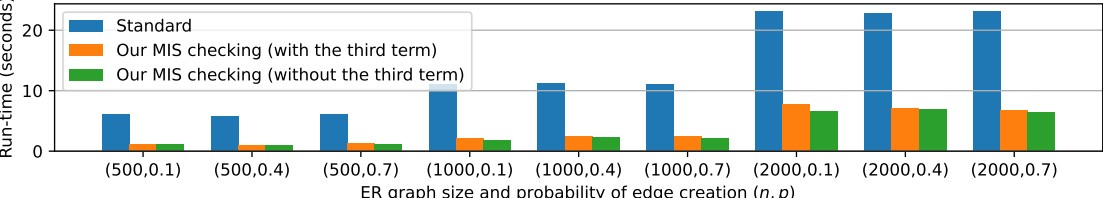

Figure 3: Average run-time results of our MIS checking vs. the standard iterative approach across different graph sizes and densities. Orange and green results correspond to using the criteria in (15) and (16), respectively.

ered in recent learning-based and sampling studies). This choice of the number of edges in the GNM graph generator indicate that half of the total possible edges (w.r.t. the complete graph) exist. Results are provided in Figure 2. As observed, for dense graphs, as the graph size increases, our method requires significantly less run-time compared to all baselines, while reporting the same average MIS size (third entry of each tuple in the x-axis). For instance, when $n$ is $500$, our method requires less than 12 seconds to solve the 5 graphs, whereas other baselines require 35 minutes or more to achieve the same MIS size. For the case of $n = 2000$, our method requires less than 4 minutes whereas ReduMIS requires nearly 350 minutes. These results indicate that, unlike ReduMIS and ILP solvers, the run-time of our method scales only with the number of nodes in the graph, which is a significant improvement.

### 4.4. Impact of the Proposed MIS Checking Criterion

In this subsection, we evaluate the impact of the proposed MIS checking method on the run-time performance of the pCQO-MIS algorithm. Specifically, we execute pCQO-MIS for $T = 1000$ iterations, performing MIS checking at each iteration. The average run-time (seconds) results for 10 ER graphs, covering various graph sizes and densities, are illustrated in Figure 3, with the x-axis representing different values of $n$ (graph size) and $p$ (probability of edge creation that indicates density). We compare these results to the standard MIS checking approach, which involves iterating over all nodes to examine their neighbors, as discussed in Section 3.3.3. The results suggest that the execution time for pCQO-MIS is significantly reduced with our efficient implementation compared to the standard method as the graph order increases.

## 5. Conclusion

This paper addressed the challenging Maximum Independent Set (MIS) Problem within the domain of Combinatorial Optimization by introducing a clique-informed continuous quadratic formulation. By eliminating the need for any training data, pCQO-MIS distinguishes itself from conventional learning approaches. Utilizing momentum-based gradient descent and a parallel GPU implementation, our straight-

forward yet effective method demonstrates competitive performance compared to state-of-the-art learning, sampling, and heuristic methods. This research offers a distinctive perspective on approaching discrete optimization problems through a parameter-efficient procedure optimized from the problem structure rather than from datasets.

## Impact Statement

This work introduces a novel quadratic optimization framework, pCQO-MIS, that advances combinatorial optimization research by tackling the Maximum Independent Set (MIS) problem with enhanced scalability and performance. By leveraging a clique-informed quadratic formulation and momentum-based parallel optimization, pCQO-MIS achieves superior MIS sizes while maintaining competitive run-times. Unlike data-centric approaches, it eliminates dependency on labeled data and out-of-distribution tuning, offering robust generalization across graph instances. Furthermore, its run-time efficiency, scaling with nodes rather than edges, positions pCQO-MIS as a transformative approach for large-scale graph problems, bridging the gap between theory and practical applicability in optimization.

## Acknowledgments

JL acknowledges Defense Advances Research projects Agency (DARPA)'s Young Faculty Award (YFA) D24AP00265. CY acknowledges NSF2349670 and NSF2403135. The authors would like to thank Curie Kim and Mingju Liu from the University of Maryland, College Park, for their help in evaluating our method with the large ER graphs under time constraints. We would also like to thank Yikai Wu and Haoyu Zhao from Princeton University for insightful discussions about the optimization in pCQO-MIS, and for sharing the BA graphs. This research was developed with funding from DARPA. The views, opinions, and/or findings expressed are those of the authors and should not be interpreted as representing the official views of the Department of Defense or the U.S. Government. Distribution Statement "A" (Approved for Public Release, Distribution Unlimited).

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

# Appendix

In this appendix, we begin with detailed proofs in Appendix A, followed by a discussion on how the proposed objective corresponds to a dataless quadratic neural network in Appendix B. Appendix C presents a study on the feasibility of saddle points. We review prior MIS solvers in Appendix D, and provide additional experimental results, implementation details, and ablation studies in Appendix E.

## A. Proofs

We begin by re-stating our main optimization problem:

$$\min_{\mathbf{x}\in[0,1]^n} f(\mathbf{x}) := -\sum_{v\in V}\mathbf{x}_v + \gamma\sum_{(u,v)\in E}\mathbf{x}_v\mathbf{x}_u - \gamma'\sum_{(u,v)\in E'}\mathbf{x}_v\mathbf{x}_u = -\mathbf{e}_n^T\mathbf{x} + \frac{\gamma}{2}\mathbf{x}^T\mathbf{A}_G\mathbf{x} - \frac{\gamma'}{2}\mathbf{x}^T\mathbf{A}_{G'}\mathbf{x}. \quad (17)$$

The gradient of $f(\mathbf{x})$ in (17) is

$$\nabla_{\mathbf{x}}f(\mathbf{x}) = \mathbf{g}(\mathbf{x}) = -\mathbf{e}_n + (\gamma\mathbf{A}_G - \gamma'\mathbf{A}_{G'})\mathbf{x}, \quad (18)$$

where, for some $v \in V$, we have

$$\frac{\partial f(\mathbf{x})}{\partial\mathbf{x}_v} = -1 + \gamma\sum_{u\in\mathcal{N}(v)}\mathbf{x}_u - \gamma'\sum_{u\in\mathcal{N}'(v)}\mathbf{x}_u. \quad (19)$$

### A.1. Proof of Lemma 1

**Re-statement**: For any non-complete graph $G$, the constant hessian of $f(\mathbf{x})$ in (4), i.e., $\gamma\mathbf{A}_G - \gamma'\mathbf{A}_{G'}$, is always a non-positive-semidefinite (non-PSD) matrix.

*Proof.* The hessian, $(\gamma\mathbf{A}_G - \gamma'\mathbf{A}_{G'})$, is independent of $\mathbf{x}$. If $(\gamma\mathbf{A}_G - \gamma'\mathbf{A}_{G'})$ is PSD, then, by definition of PSD matrices, we must have

$$\mathbf{x}^T(\gamma\mathbf{A}_G - \gamma'\mathbf{A}_{G'})\mathbf{x} \geq 0, \forall\mathbf{x}\in[0,1]^n, \quad (20)$$

which is not possible as for any $\mathbf{x}_0$ that corresponds to a MaxIS, we have $\mathbf{x}_0^T(\gamma\mathbf{A}_G)\mathbf{x}_0 = 0$ (no edges in MaxIS w.r.t. $G$) and $\gamma'\mathbf{x}_0^T\mathbf{A}_{G'}\mathbf{x}_0 < 0$ (a MaxIS in $G$ is a maximal clique in $G'$). □

### A.2. Proof of Theorem 1

**Re-statement**: Given an arbitrary graph $G = (V, E)$ and its corresponding formulation in (17), suppose the size of any MIS of $G$ is $k$. Then, $\gamma \geq 1 + \gamma'k$ is necessary and sufficient for all MaxIS vectors to be local minimizers of (17) for arbitrary graphs.

*Proof.* Let $\mathcal{I}$ be a MaxIS. Define the vector $\mathbf{x}^{\mathcal{I}}$ such that it contains 1's at positions corresponding to the nodes in the set $S$, and 0's at all other positions. For any MaxIS to be a local minimizer of (17), it is sufficient and necessary to require that

$$\frac{\partial f(\mathbf{x})}{\partial\mathbf{x}_v} \geq 0, \quad \forall v \notin \mathcal{I} \text{ and} \quad (21)$$

$$\frac{\partial f(\mathbf{x})}{\partial\mathbf{x}_v} \leq 0, \quad \forall v \in \mathcal{I}. \quad (22)$$

Here, $\mathbf{x}_v$ is the element of $\mathbf{x}$ at the position corresponding to node $v$. Equation (21) is derived because if $v \notin \mathcal{I}$, then $\mathbf{x}_v^{\mathcal{I}} = 0$ (by the definition of $\mathbf{x}^{\mathcal{I}}$) so it is at the left boundary of the interval $[0, 1]$. For the left boundary point to be a local minimizer, it requires the derivative to be non-negative (i.e., moving towards the right only increases the objective). Similarly, when $v \in \mathcal{I}$, $\mathbf{x}_v^{\mathcal{I}} = 1$, is at the right boundary for (22), at which the derivative should be non-positive.

The derivative of $f$ computed in (19) can be rewritten as

$$\frac{\partial f(\mathbf{x})}{\partial \mathbf{x}_v} = -1 + \gamma m_v - \gamma' \ell_v, \quad \forall v \notin \mathcal{I}, \tag{23}$$

where

$$m_v := |\{u \in \mathcal{N}(v) \cap \mathcal{I}\}| \, , \tag{24}$$

is the number of neighbors of $v$ in $\mathcal{I}$ and

$$\ell_v := |\{u \in \mathcal{N}'(v) \cap \mathcal{I}\}| \, , \tag{25}$$

is the number of non-neighbors of $v$ in $\mathcal{I}$, Here, $\mathcal{N}'(v) = \{u : (u,v) \in E'\}$.

By this definition, we immediately have $1 \leq m_v \leq |\mathcal{I}|$ and $0 \leq \ell_v \leq |\mathcal{I}|$, where the upper and lower bounds for $m_v$ and $\ell_v$ are all attainable by some special graphs. Note that the lower bound of $m_v$ is 1, and that is due the fact that $\mathcal{I}$ is a MaxIS, so any other node (say $v$) will have at least 1 edge connected to a node in $\mathcal{I}$.

Plugging (23) into (21), we obtain

$$\gamma \geq \frac{1 + \gamma' \ell_v}{m_v} \, . \tag{26}$$

Since we're seeking a universal $\gamma$ for all the graphs, we must set $m_v$ to its lowest possible value, 1, and $\ell_v$ to its highest possible value $k$ (both are attainable by some graphs), and still requires $\gamma$ to satisfy (26). This means it is necessary and sufficient to require $\gamma \geq 1 + \gamma' k$. In addition, (22) is satisfied unconditionally and therefore does not impose any extra condition on $\gamma$. $\qquad \square$

## A.3. Proof of Lemma 2

**Re-statement**: All local minimizers of (17) are binary vectors.

*Proof.* Let $\mathbf{x}^*$ be any local minimizer of (17). If all the coordinates of $\mathbf{x}^*$ are either 0 or 1, then $\mathbf{x}^*$ is binary and the proof is complete, otherwise, at least one coordinate of $\mathbf{x}^*$ is in the interior $(0,1)$ and we aim to prove that this is not possible (i.e. such a non-binary $\mathbf{x}^*$ cannot exist as a minimizer) by contradiction. We assume the non-binary $\mathbf{x}^*$ exists, and denote the set of non-binary coordinates as

$$J := \{j : \mathbf{x}_j^* \in (0,1)\} \, . \tag{27}$$

Since $\mathbf{x}^*$ is non-binary, $J \neq \emptyset$. Since the objective function $f(\mathbf{x})$ of (17) is twice differentiable with respect to all $\mathbf{x}_j$ with $\mathbf{x}_j \in (0,1)$, then a necessary condition for $\mathbf{x}^*$ to be a local minimizer is that

$$\nabla f(\mathbf{x}^*)\big|_J = 0, \quad \nabla^2 f(\mathbf{x}^*)\big|_J \succeq 0,$$

where $\nabla f(\mathbf{x}^*)\big|_J$ is the vector $\nabla f(\mathbf{x}^*)$ restricted to the index set $J$, and $\nabla^2 f(\mathbf{x}^*)\big|_J$ is the matrix $\nabla^2 f(\mathbf{x}^*)$ whose row and column indices are both restricted to the set $J$.

However, the second necessary condition $\nabla^2 f(\mathbf{x}^*)\big|_J \succeq 0$ cannot hold. Because if it does, then we must have $\mathrm{tr}(\nabla^2 f(\mathbf{x}^*)\big|_J) > 0$ (the trace cannot strictly equal to 0 as $\nabla^2 f(\mathbf{x}^*)\big|_J = \mathbf{I}_J(\gamma \mathbf{A}_G - \gamma' \mathbf{A}_{G'})\mathbf{I}_J^T \neq 0$ where $\mathbf{I}_J$ denotes the identity matrix with row indices restricted to the index set $J$). However, on the other hand, we have

$$\mathrm{tr}(\nabla^2 f(\mathbf{x}^*)\big|_J) = \mathrm{tr}(\mathbf{I}_J(\gamma \mathbf{A}_G - \gamma' \mathbf{A}_{G'})\mathbf{I}_J^T) = 0 \, ,$$

as the diagonal entries of $\mathbf{A}_G$ and $\mathbf{A}_{G'}$ are all 0, which leads to a contradiction. $\qquad \square$

## A.4. Proof of Theorem 2

**Re-statement**: Given graph $G = (V, E)$ and set $\gamma \geq 1 + \gamma' \Delta(G')$, all local minimizers of (17) correspond to a MaxIS in $G$.

*Proof.* By lemma 2, we only consider binary vectors as local minimizers. With this, we first prove that all local minimizers are Independent Sets (ISs). Then, we show that any IS, that is not a maximal IS, is not a local minimizer.

Here, we show that any local minimizer is an IS. By contradiction, assume that vector $\mathbf{x}$, by which $\mathbf{x}_v = \mathbf{x}_w = 1$ such that $(v, w) \in E$ (a binary vector with an edge in $G$), is a local minimizer. Since $\mathbf{x}_v = 1$ is at the right boundary of the interval $[0, 1]$, for it to be a local minimizer, we must have $\frac{\partial f}{\partial \mathbf{x}_v} \leq 0$. Together with (19), this implies

$$-1 + \gamma \sum_{u \in \mathcal{N}(v)} \mathbf{x}_u - \gamma' \sum_{u \in \mathcal{N}'(v)} \mathbf{x}_u \leq 0 \,. \tag{28}$$

Re-arranging (28) yields to

$$\gamma \sum_{u \in \mathcal{N}(v)} \mathbf{x}_u \leq 1 + \gamma' \sum_{u \in \mathcal{N}'(v)} \mathbf{x}_u \,. \tag{29}$$

Given that $\gamma \geq 1 + \gamma' \Delta(G')$, the condition in (29) can not be satisfied even if the LHS attains its minimum value (which is $\gamma n$) and the RHS attains a maximum value. The maximum possible value of the RHS is $1 + \mathrm{d}'(v) = n - \mathrm{d}(v)$, where $\mathrm{d}'(v)$ is the degree of node $v$ in $G'$, and the maximum possible value of $\mathrm{d}'(v)$ is $\Delta(G')$. This means that when an edge exists in $\mathbf{x}$, it can not be a fixed point. Thus, only ISs are local minimizers.

Here, we show that Independent Sets that are not maximal are not local minimizers. Define vector $\mathbf{x} \in \{0, 1\}^n$ that corresponds to an IS $\mathcal{I}(\mathbf{x})$. This means that there exists a node $u \in V$ that is not in the IS and is not in the neighbor set of all nodes in the IS. Formally, if there exists $u \notin \mathcal{I}(\mathbf{x})$ such that $\forall w \in \mathcal{I}(\mathbf{x}), u \notin \mathcal{N}(w)$, then $\mathcal{I}(\mathbf{x})$ is an IS, not a maximal IS. Note that such an $\mathbf{x}$ satisfies $\mathbf{x}_u = 0$ and

$$\frac{\partial f}{\partial \mathbf{x}_v} = -1 + \gamma \sum_{u \in \mathcal{N}(v)} \mathbf{x}_u - \gamma' \sum_{u \in \mathcal{N}'(v)} \mathbf{x}_u = -1 - \gamma' \sum_{u \in \mathcal{N}'(v)} \mathbf{x}_u < 0 \,, \tag{30}$$

which implies that increasing $\mathbf{x}_u$ can further decrease the function value, contradicting to $\mathbf{x}$ being a local minimizer. In (30), the second summation is 0 as $\mathcal{N}(v) \cap \mathcal{I}(\mathbf{x}) = \emptyset$, which results in $-(1 + \gamma' \sum_{u \in \mathcal{N}'(v)} \mathbf{x}_u)$ that is always negative. Thus, any binary vector that corresponds to an IS that is not maximal is not a local minimizer. $\square$

### A.5. Proof of Theorem 3

**Re-statement**: For any graph $G$, assume that there exists a point $\mathbf{x}'$ such that $\nabla_{\mathbf{x}} f(\mathbf{x}') = \mathbf{0}$, i.e., $\mathbf{x}' = (\gamma \mathbf{A}_G - \gamma' \mathbf{A}_{G'})^{-1} \mathbf{e}_n$. Then, $\mathbf{x}'$ is not a local minimizer of (17) and therefore does not correspond to a MaxIS.

*Proof.* By Lemma 2, we know that all local minimizers are binary. By contradiction, assume that $\mathbf{x}'$ is a binary local minimizer. Then, the system of equations $(\gamma \mathbf{A}_G - \gamma' \mathbf{A}_{G'}) \mathbf{x}' = \mathbf{e}_n$ implies that, for all $v \in V$, the following equality must be satisfied.

$$\gamma \sum_{u \in \mathcal{N}(v)} \mathbf{x}_u - \gamma' \sum_{u \in \mathcal{N}'(v)} \mathbf{x}_u = 1 \,. \tag{31}$$

If $\mathbf{x}'$ is binary and corresponds to a MaxIS in the graph, then the first term of (31) is always 0, which reduces (31) to

$$-\gamma' \sum_{u \in \mathcal{N}'(v)} \mathbf{x}_u = 1 \,. \tag{32}$$

Eq.(32) is an equality that can not be satisfied as $\mathbf{x}'_v \geq 0, \forall v \in V$ and $\gamma' \geq 1$. Thus, $\mathbf{x}'$ is not a local minimizer. $\square$

## B. Connection to dataless Quadratic Neural Networks

Our proposed objective in (4) corresponds to a dataless quadratic neural networks (dQNN), as illustrated in Figure 4 (*right*). Here, the dQNN comprises two fully connected layers. The initial activation-free layer encodes information about the nodes (top $n = 5$ connections), edges of $G$ (middle $m = 4$ connections), and edges of $G'$ (bottom $m' = 6$ connections), all without a bias vector. The subsequent fully connected layer is an activation-free layer performing a vector dot-product between the fixed weight vector (with $-1$ corresponding to the nodes and edges of $G'$ and the edges-penalty parameter $\gamma$), and the output of the first layer.

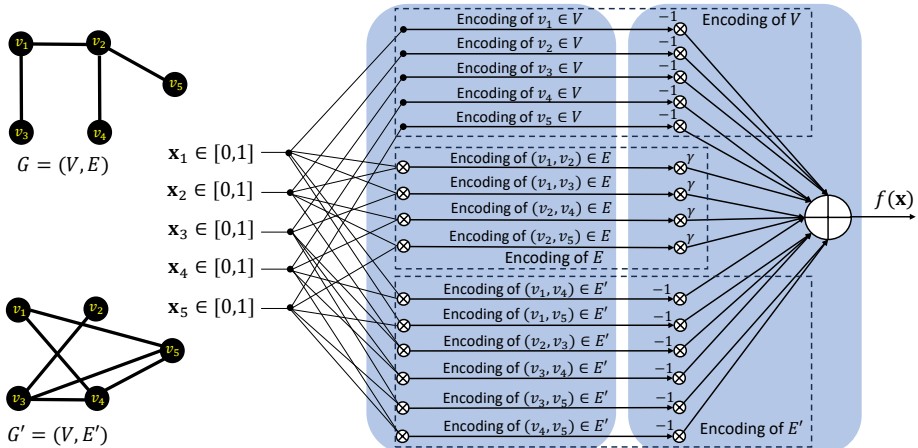

Figure 4: Graph $G$ (*left*) and its corresponding dataless quadratic neural network (*right*).

## C. Empirical Observations on the Non-Extremal Stationary Point $\mathbf{x}'$

In this section, we empirically demonstrate how the non-extremal stationary point $\mathbf{x}'$, analyzed in Theorem 3, varies with the type of graph. Specifically, we aim to show that, for many types of graphs, this saddle point is outside the box constraints, depending on the graph connectivity. To this end, we consider GNM and ER graphs with different densities, as well as small and large graphs from the SATLIB dataset.

In Figure 5, we obtain $\mathbf{x}' = (\gamma \mathbf{A}_G - \gamma' \mathbf{A}_{G'})^{-1} \mathbf{e}_n$ with $\gamma = n$ and $\gamma' = 1$ for every considered graph. Each subplot in Figure 5 shows the values of $\mathbf{x}'_v$ (y-axis) for every node $v \in V$ (x-axis), with the title specifies the graph used.

As observed, among all the graphs, only the very-high-density GNM graph (with results shown inside the dashed box in Figure 5) has $\mathbf{x}' \in [0, 1]^n$ (i.e., within the box-constraints of (4)). Note that this graph was generated with $m = 4945$ where the total number of possible edges in the complete graph with $n = 100$ is 4950 edges.

For all other graphs, we have $\mathbf{x}' \notin [0, 1]^n$, as indicated by the values strictly below 0. This means that by applying the projection in (12), $\mathbf{x}'$ is infeasible.

## D. Related Work

**1) Exact and Heuristic Solvers:** Exact approaches for NP-hard problems typically rely on branch-and-bound global optimization techniques. However, exact approaches suffer from poor scalability, which limits their uses in large MIS problems (Dai et al., 2016). This limitation has spurred the development of efficient approximation algorithms and heuristics. For instance, the well-known NetworkX library (Hagberg et al., 2008) implements a heuristic procedure for solving the MIS problem (Boppana & Halldórsson, 1992). These polynomial-time heuristics often incorporate a mix of sub-procedures, including greedy algorithms, local search sub-routines, and genetic algorithms (Williamson & Shmoys, 2011). However, such heuristics generally cannot theoretically guarantee that the resulting solution is within a small factor of optimality. In fact, inapproximability results have been established for the MIS problem (Berman & Schnitger, 1992).

Among existing MIS heuristics, ReduMIS (Lamm et al., 2016) has emerged as the leading approach. The ReduMIS framework contains two primary components: (*i*) an iterative application of various graph reduction techniques (e.g., the linear programming (LP) reduction method in (Nemhauser & Trotter, 1975)) with a stopping rule based on the non-applicability of these techniques; and (*ii*) an evolutionary algorithm. The ReduMIS algorithm initiates with a pool of independent sets and evolves them through multiple rounds. In each round, a selection procedure identifies favorable nodes by executing graph partitioning, which clusters the graph nodes into disjoint clusters and separators to enhance the solution. In contrast, our pCQO-MIS approach does *not* require such complex algorithmic operations (e.g., solution combination operation, community detection, and local search algorithms for solution improvement (Andrade et al., 2012)) as used in ReduMIS. More importantly, ReduMIS and ILP solvers scale with the number of nodes and the number of edges (which constraints their application on highly dense graphs), whereas pCQO-MIS only scales w.r.t. the number nodes, as demonstrated in our experimental results (Section 4.3).

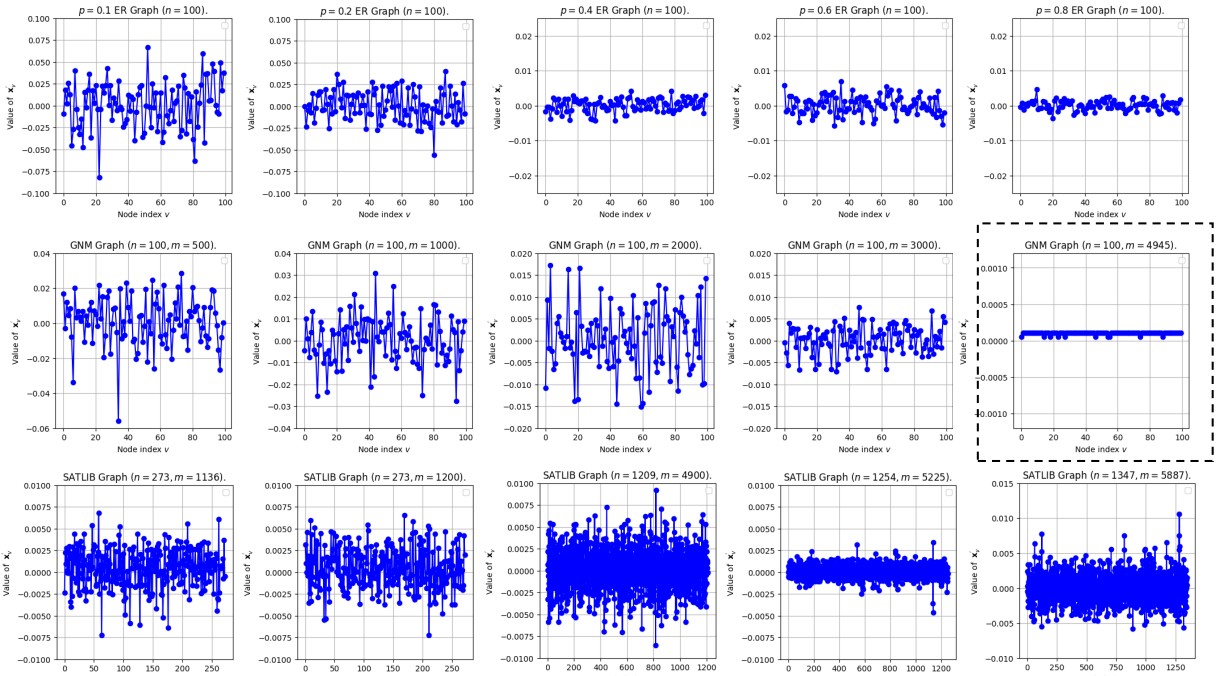

Figure 5: Values of the non-extremal stationary point $\mathbf{x}'$ (y-axis) w.r.t. every node $v \in V$ (x-axis) across different ER and GNM graphs as well as small and large SATLIB graphs, as indicated by the title of each subplot. Among all the considered graphs, only the high-density GNM graph, indicated by the dashed square, has $\mathbf{x}' \in [0,1]^{100}$.

**2) Data-Driven Learning-Based Solvers:** Data-driven approaches for the MIS problem can be classified into SL and RL methods. These methods depend on neural networks trained to fit the distribution over (un)labeled training graphs.

A notable SL method is proposed in (Li et al., 2018), which combines several components including graph reductions (Lamm et al., 2016), Graph Convolutional Networks (GCN) (Defferrard et al., 2016), guided tree search, and the solution improvement local search algorithm (Andrade et al., 2012). The GCN is trained on SATLIB graphs using their solutions as ground truth labels, enabling the learning of probability maps for the inclusion of each vertex in the optimal solution. Then, a subset of ReduMIS subroutines is used to improve their solution. While the work in (Li et al., 2018) reported on-par results to ReduMIS, it was later shown by (Böther et al., 2022) that setting the GCN parameters to random values performs similarly to using the trained GCN network.

Recently, DIFUSCO was introduced in (Sun & Yang, 2023), an approach that integrates Graph Neural Networks (GNNs) with diffusion models (Ho et al., 2020) to create a graph-based diffusion denoiser. DIFUSCO formulates the MIS problem in the discrete domain and trains a diffusion model to improve a single or a pool of solutions.

RL-based methods have achieved more success in solving the MIS problem when compared to SL methods. In (Dai et al., 2017), a Deep Q-Network (DQN) is combined with graph embeddings, facilitating the discrimination of vertices based on their influence on the solution and ensuring scalability to larger instances. Meanwhile, the study presented in (Ahn et al., 2020) introduces the Learning What to Defer (LwD) method, an unsupervised deep RL solver resembling tree search, where vertices are iteratively assigned to the independent set. Their model is trained using Proximal Policy Optimization (PPO) (Schulman et al., 2017).

The work in (Qiu et al., 2022) introduces DIMES, which combines a compact continuous space to parameterize the distribution of potential solutions and a meta-learning framework to facilitate the effective initialization of model parameters during the fine-tuning stage that is required for each graph.

It is worth noting that the majority of SL and RL methods are *data-dependent* in the sense that they often require the training of a separate network for each dataset of graphs. These data-dependent methods exhibit limited *generalization* performance when applied to out-of-distribution graph data. This weak generalization stems from the need to train a different network for each graph dataset (see columns 3 and 6 in Table 1). An example of the weak generalization of DIFUSCO is given in

Appendix E.8. In contrast, our approach differs from SL- and RL-based methods in that it does not rely on any training datasets. Instead, our method utilizes a simple yet effective *graph-encoded* continuous objective function, which is defined solely in terms of the connectivity of a given graph.

**3) Dataless Differentiable Methods:** The method in (Alkhouri et al., 2022) introduced dataless neural networks (dNNs) tailored for the MIS problem. Notably, their method operates without the need for training data and relies on $n$ trainable parameters. Their proposed method uses a ReLU-based continuous objective to solve the MIS problem. However, for scaling up, graph partitioning and local search algorithms were employed.

The work in (Schuetz et al., 2022) introduced Physics-Inspired Graph Neural Network (PI-GNN), a dataless approach for solving COPs that optimizes the parameters of a GNN over one graph using a continuous relaxation of (3) with box-constraints. However, only $d$-regular graphs were used for evaluation. Multiple studies followed PI-GNN including the work in (Ichikawa, 2024).

**4) Discrete Sampling Solvers:** In recent studies, researchers have explored the integration of energy-based models with parallel implementations of simulated annealing to address combinatorial optimization problems (Goshvadi et al., 2024) without relying on any training data. For example, in tackling the MIS problem, the work in (Sun et al., 2023) proposed a solver that combines (*i*) Path Auxiliary Sampling (PAS) (Sun et al., 2021) and (*ii*) the QUBO formulation in (3). However, unlike pCQO-MIS, these approaches entail prolonged sequential run-time and require fine-tuning of several hyperparameters. Moreover, the energy models utilized in this method for addressing the MIS problem may generate binary vectors that violate the "no edges" constraint inherent to the MIS problem. Consequently, a post-processing procedure becomes necessary.

### D.1. Requirements Comparison with Baselines

In Table 2, we provide an overview comparison of the number of trainable parameters, hyper-parameters, and additional techniques needed for each baseline. ReduMIS depends on a large set of graph reductions (see Section 3.1 in (Lamm et al., 2016)) and graph clustering, which is used for solution improvement.

| Method | Size | Hyper-Parameters | Additional Techniques/Procedures |
|---|---|---|---|
| ReduMIS | $n$ variables | N/A | Many graph reductions, and graph clustering |
| Gurobi | $n$ variables | N/A | N/A |
| CP-SAT | $n$ variables | N/A | N/A |
| GCN | $\gg n$ trainable parameters | Many as it is learning-based | Tree Search |
| LwD | $\gg n$ trainable parameters | Many as it is learning-based | Entropy Regularization |
| DIMES | $\gg n$ trainable parameters | Many as it is learning-based | Tree Search or Sampling Decoding |
| DIFUSCO | $\gg n$ trainable parameters | Many as it is learning-based | Greedy Decoding or Sampling Decoding |
| iSCO | $n$ variables | Temperature, Sampler, Chain length | Post Processing for Correction |
| pCQO-MIS | $n$ trainable parameters | $\alpha, \beta, \gamma, \gamma', T$, and $\eta$ | Degree-based Parallel Initializations |

Table 2: Requirements comparison with baselines. For the ILPs (Gurobi and CP-SAT), trainable parameters correspond to $n$ binary decision variables. ReduMIS is not an optimization method. However, they use $n$ binary variables, one for each node.

For learning-based methods, although they attempt to 'fit' a distribution over training graphs, the parameters of a neural network architecture are optimized during training. This architecture is typically much larger than the number of input coordinates ($\gg n$). For instance, the network used in DIFUSCO consists of 12 layers, each with 5 trainable weight matrices. Each weight matrix is of size $256 \times 256$, resulting in 3932160 trainable parameters for the SATLIB dataset (which consists of graphs with at most 1347 nodes). Moreover, this dependence on training a NN introduces several hyper-parameters such as the number of layers, size of layers, choice of activation functions, etc.

It's important to note that the choice of the sampler in iSCO introduces additional hyper-parameters. For instance, the PAS sampler (Sun et al., 2021) used in iSCO depends on the choice of the neighborhood function, a prior on the path length, and the choice of the probability of acceptance.

# E. Additional Experiments

## E.1. Results using DIMACS Graphs

In this section, we evaluate our proposed algorithm using graph instances from the DIMACS dataset. These graph instances have known optimal solutions as listed in the recent MC survey paper (Marino et al., 2024). The DIMACS benchmark is part of the second DIMACS Implementation Challenge (Johnson & Trick, 1996), which focused on problems related to Clique, Satisfiability, and Graph Coloring. The benchmark contains a variety of graphs derived from coding theory, and fault diagnosis, among others.

As observed, we were able to solve 49 out of the 61 DIMACS graphs we tested within a 30-second time budget per graph, while ReduMIS was able to solve 58 in the same amount of time.

For our method, we use the following set of hyper-parameters: $\alpha = 0.01, \beta = 0.3, \gamma = 500, \gamma' = 1, \eta = 2.25, T = 500$. We emphasize that these graphs are very diverse (in terms of both order and density as indicated in columns 2 and 4) and using one set of hyper-parameters indicate that our method may not be highly sensitive in terms of finding feasible solutions. This also indicates that if we perform a per graph hyper-parameters tuning, our method has the potential of reporting improved results.

| Graph Name | $n$ | $m$ | Density | Optimal | pCQO-MIS | ReduMIS |
|---|---|---|---|---|---|---|
| c-fat500-1 | 500 | 120291 | 0.9600 | 14 | 14 | 14 |
| c-fat500-2 | 500 | 115611 | 0.9267 | 26 | 26 | 26 |
| c-fat200-1 | 200 | 18366 | 0.9229 | 12 | 12 | 12 |
| c-fat200-2 | 200 | 16665 | 0.8374 | 24 | 24 | 24 |
| c-fat500-5 | 500 | 101559 | 0.8141 | 64 | 64 | 64 |
| p_hat300-1 | 300 | 33917 | 0.7562 | 8 | 8 | 8 |
| p_hat1000-1 | 1000 | 377247 | 0.7552 | 10 | 10 | 10 |
| p_hat700-1 | 700 | 183651 | 0.7507 | 11 | 11 | 11 |
| p_hat500-1 | 500 | 93181 | 0.7469 | 9 | 9 | 9 |
| p_hat1500-1 | 1500 | 839327 | 0.7466 | 12 | **11** | 12 |
| hamming6-4 | 64 | 1312 | 0.6508 | 4 | 4 | 4 |
| c-fat500-10 | 500 | 78123 | 0.6262 | 126 | 126 | 126 |
| c-fat200-5 | 200 | 11427 | 0.5742 | 58 | 58 | 58 |
| p_hat300-2 | 300 | 22922 | 0.5111 | 25 | 25 | 25 |
| p_hat1000-2 | 1000 | 254701 | 0.5099 | 46 | 46 | 46 |
| brock200_2 | 200 | 10024 | 0.5037 | 12 | **11** | 12 |
| p_hat700-2 | 700 | 122922 | 0.5024 | 44 | 44 | 44 |
| DSJC1000_5 | 1000 | 249674 | 0.4998 | 15 | 15 | 15 |
| C2000.5 | 2000 | 999164 | 0.4998 | 16 | **15** | 16 |
| sanr400_0.5 | 400 | 39816 | 0.4989 | 13 | 13 | 13 |
| DSJC500_5 | 500 | 62126 | 0.4980 | 13 | 13 | 13 |
| p_hat500-2 | 500 | 61804 | 0.4954 | 36 | 36 | 36 |
| p_hat1500-2 | 1500 | 555290 | 0.4939 | 65 | 65 | 65 |
| johnson8-2-4 | 28 | 168 | 0.4444 | 4 | 4 | 4 |
| brock200_3 | 200 | 7852 | 0.3946 | 15 | **14** | 15 |
| hamming8-4 | 256 | 11776 | 0.3608 | 16 | 16 | 16 |
| keller4 | 171 | 5100 | 0.3509 | 11 | 11 | 11 |
| brock800_1 | 800 | 112095 | 0.3507 | 23 | **20** | **21** |
| brock200_4 | 200 | 6811 | 0.3423 | 17 | 16 | 17 |
| sanr200_0.7 | 200 | 6032 | 0.3031 | 18 | 18 | 18 |
| san200_0.7_1 | 200 | 5970 | 0.3000 | 30 | 30 | 30 |
| sanr400_0.7 | 400 | 23931 | 0.2999 | 21 | 21 | 21 |
| p_hat1000-3 | 1000 | 127754 | 0.2558 | 68 | **67** | 68 |

*Continued on next page*

| Graph Name | $n$ | $m$ | Density | Optimal | pCQO-MIS | ReduMIS |
|---|---|---|---|---|---|---|
| p_hat300-3 | 300 | 11460 | 0.2555 | 36 | 36 | 36 |
| brock200_1 | 200 | 5066 | 0.2546 | 21 | 21 | 21 |
| p_hat700-3 | 700 | 61640 | 0.2520 | 62 | 62 | 62 |
| brock400_1 | 400 | 20077 | 0.2516 | 27 | **25** | **25** |
| p_hat500-3 | 500 | 30950 | 0.2481 | 50 | 50 | 50 |
| p_hat1500-3 | 1500 | 277006 | 0.2464 | 94 | **93** | 94 |
| johnson16-2-4 | 120 | 1680 | 0.2353 | 8 | 8 | 8 |
| johnson8-4-4 | 70 | 560 | 0.2319 | 14 | 14 | 14 |
| hamming10-4 | 1024 | 89600 | 0.1711 | 40 | 40 | 40 |
| johnson32-2-4 | 496 | 14880 | 0.1212 | 16 | 16 | 16 |
| sanr200_0.9 | 200 | 2037 | 0.1024 | 42 | 42 | 42 |
| C125.9 | 125 | 787 | 0.1015 | 34 | 34 | 34 |
| C250.9 | 250 | 3141 | 0.1009 | 44 | 44 | 44 |
| gen400_p0.9_75 | 400 | 7980 | 0.1000 | 75 | 75 | 75 |
| gen400_p0.9_55 | 400 | 7980 | 0.1000 | 55 | **52** | 55 |
| gen200_p0.9_44 | 200 | 1990 | 0.1000 | 44 | **42** | 44 |
| gen200_p0.9_55 | 200 | 1990 | 0.1000 | 55 | 55 | 55 |
| san200_0.9_2 | 200 | 1990 | 0.1000 | 60 | 60 | 60 |
| san400_0.9_1 | 400 | 7980 | 0.1000 | 100 | 100 | 100 |
| san200_0.9_1 | 200 | 1990 | 0.1000 | 70 | 70 | 70 |
| san200_0.9_3 | 200 | 1990 | 0.1000 | 44 | 44 | 44 |
| gen400_p0.9_65 | 400 | 7980 | 0.1000 | 65 | 65 | 65 |
| C500.9 | 500 | 12418 | 0.0995 | 57 | **56** | 57 |
| C1000.9 | 1000 | 49421 | 0.0989 | 68 | **65** | **67** |
| hamming6-2 | 64 | 192 | 0.0952 | 32 | 32 | 32 |
| MANN_a9 | 45 | 72 | 0.0727 | 16 | 16 | 16 |
| hamming8-2 | 256 | 1024 | 0.0314 | 128 | 128 | 128 |
| hamming10-2 | 1024 | 5120 | 0.0098 | 512 | 512 | 512 |

Table 3: Performance of pCQO-MIS on the DIMACS graphs dataset as compared to the known optimal solution (column 5) and SOTA heuristic ReduMIS (column 7). Graphs are ordered based on the graph density $\frac{2m}{n(n-1)}$ (column 4). For our method, the time limit is 30 seconds per graph. Bold results indicate the cases where pCQO-MIS or ReduMIS don't achieve the optimal.

## E.2. Results of Large Random ER Graphs Under Time Constraints

In this subsection, we compare our method with Gurobi and ReduMIS using 10 ER graphs with $n = 3000$ and $p = \{0.1, 0.2, 0.3, 0.4, 0.5, 0.6, 0.7\}$ with time budget of 30 seconds using the following machine: CPU Intel(R) Xeon(R) Gold 6418H and GPU NVIDIA RTX A6000.

| Method | Average MIS Size at different $p$ | | | | | | |
|---|---|---|---|---|---|---|---|
| | $p = 0.1$ | $p = 0.2$ | $p = 0.3$ | $p = 0.4$ | $p = 0.5$ | $p = 0.6$ | $p = 0.7$ |
| ReduMIS | 61.5 | – | – | – | – | – | – |
| Gurobi | 55.6 | 29.1 | 20.3 | 14.3 | 10.8 | 8.8 | 7.1 |
| **pCQO-MIS** (Ours) | **76.5** | **39.8** | **25.2** | **18.5** | **14.3** | **11.5** | **9** |

Table 4: Evaluation of pCQO-MIS vs. the ReduMIS and Gurobi with a time budget of 30 seconds using ER graphs with $n = 3000$ and different probability of edge creation, i.e., $p$. This $p$ approximately indicates the density in the graph. This means that the number of edges is from 449850 (for $p = 0.1$) to 3148950 (for $p = 0.7$).

Results are given in Table 4. As observed, under a time budget of 30 seconds, out method outperforms the ILP solver and ReduMIS. The hyper-parameters tuning was done using one graph for every $p$ and as recommended in Appendix E.9.1. For example, for $p = 0.1$, we used one graph out of the 10 and performed the quick grid search, then used the parameters for the

remaining 9.

### E.3. Results of BA Graphs from (Wu et al., 2025)

In this subsection, we report results on Barabási–Albert (BA) graphs corresponding to those used in Table 2 of (Wu et al., 2025). These graphs were generated using the NetworkX library and are parameterized by $n$ and $q$, where $n$ denotes the number of nodes and $q$ denotes the number of edges attached from a new node to existing nodes (Barabási & Albert, 1999). We evaluate graphs with $n \in \{100, 300, 1000\}$. The results from ReduMIS and OnlineMIS are taken directly from (Wu et al., 2025), **while results for our method were obtained using a total runtime of 33.7 minutes**, which includes hyperparameter tuning. **This differs from (Wu et al., 2025)'s setup, which used a time limit of up to 96 hours.** For additional details on their experiment setup and hardware, see the caption of Table 2 in (Wu et al., 2025).

Results are presented in Table 5. We match ReduMIS exactly in six cases, and report a close result in another. However, in two cases, our method underperforms with a difference of more than three nodes.

| $n$ | $q$ | OnlineMIS | ReduMIS | **Average MIS Size pCQO-MIS** (Ours) |
|---|---|---|---|---|
| 100 | 5 | 39.5 | 39.5 | 39.5 |
| 100 | 15 | 21.63 | 21.63 | 21.63 |
| 300 | 5 | 123.13 | 123.13 | 123.13 |
| 300 | 15 | 71.38 | 71.38 | 71.38 |
| 300 | 50 | 49.88 | 50 | 50 |
| 1000 | 5 | 417.13 | 417.13 | 416.625 |
| 1000 | 15 | 245 | 246.38 | 241.875 |
| 1000 | 50 | 115.75 | 116.88 | 111.75 |
| 1000 | 150 | 150 | 150 | 150 |

Table 5: Evaluation of pCQO-MIS vs. ReduMIS using a set of the BA graphs in (Wu et al., 2025). OnlineMIS is an accelerated version of ReduMIS, where a fewer number of graphs reductions are used after performing local search. Results of ReduMIS and OnlineMIS are as reported in (Wu et al., 2025).

In our experiment, we used $\gamma' = 1$, $T = 250$, and $\beta = 0.9$. For the remaining hyperparameters, we perform a grid search over $\alpha \in \{0.01, 0.001, 0.0001, 0.00001\}$ and $\gamma \in \{100, 200, 500, 750\}$ for each graph and report the best result.

### E.4. Impact of the Adopted Momentum-based Gradient Descent Optimizer

Extremal stationary points may be rare and depend of the graph connectivity as was discussed in Appendix A.5. However, our use of MGD is not solely motivated by the need to escape these unwanted points when they exist. It is also driven by the empirical observation that, when starting from the same initial point, MGD converges to minimizers with larger MaxIS values while avoiding the overshooting observed with vanilla GD. Also, momentum is generally used to accelerate convergence of GD. To support our claim that MGD converges to better minima compared to GD, we conduct the following experiment: We use 5 ER graphs with $n = 100$ and $p \in \{0.3, 0.6\}$ (probability of edge creation) and run GD vs. MGD, using the same $\gamma, \gamma', \alpha$ and the initializations. Table 6 shows the results. As observed, on average, MGD converges to larger MIS. Furthermore, MGD avoids the all 0's which is the case of overshooting in GD.

| | $p = 0.3$ | | $p = 0.6$ | |
|---|---|---|---|---|
| **Step size $\alpha$** | **GD Avg. MIS** | **MGD Avg. MIS** | **GD Avg. MIS** | **MGD Avg. MIS** |
| 0.0001 | 11.2 | 12.9 | 6.7 | 6.9 |
| 0.0002 | 11.7 | 12.8 | 0.0 | 6.3 |

Table 6: Comparison of average MIS sizes for different step sizes $\alpha$ using GD and MGD across different densities (as indicated by $p = 0.3$ and $p = 0.6$). For both cases, we use $\gamma = n$ and $\gamma' = 1$.

### E.5. Ablation Study on the Clique Term in pCQO-MIS

In pCQO-MIS, the clique term is introduced to (*i*) encourage the optimizer to select two nodes connected by an edge in the complement graph, leveraging the duality between the clique and MIS problems, and (*ii*) to discourage sparsity in the solution given the $\ell_1$ norm in (6). This is our motivation and intuition.

The improvements are observed empirically in terms of enhancing stability, preventing overshooting, and leading to better minima.

Tables 7 and 8 compare the cases with and without the clique term (i.e., $\gamma' = 7$ vs. $\gamma' = 0$) over the ER dataset used in Table 1. The results are presented as the average MIS size (Table 7) and the number of steps for first solution (Table 8) in the format "without–with" and are obtained across different values of $\alpha$ (step size) and $\gamma$ (edge penalty parameter). The results are reported after optimizing 50 batches of initializations for each unique set of hyper-parameters. We note that the range of $\gamma$ is selected based on the criterion in Theorem 2.

| Step size $\alpha$ | $\gamma = 350.0$ | $\gamma = 450.0$ | $\gamma = 525.0$ | $\gamma = 600.0$ |
|---|---|---|---|---|
| 4e-06 | 0.0 - 39.07 | 0.0 - 39.18 | 0.0 - 40.27 | 0.0 - **42.59** |
| 9e-06 | 0.0 - **44.51** | 0.0 - **44.21** | 0.0 - **43.87** | 0.0 - **43.57** |
| 4e-05 | 0.0 - **41.52** | 0.0 - **41.21** | 0.0 - **40.99** | 0.0 - **40.69** |
| 9e-05 | 0.0 - **40.70** | 0.0 - **40.61** | 0.0 - **40.54** | 0.0 - **40.60** |
| 0.0004 | 40.50 - **41.14** | 40.50 - **41.00** | 40.35 - **40.94** | 40.39 - **40.71** |
| 0.0009 | 40.34 - **41.24** | 40.42 - **41.16** | 40.44 - **40.91** | 40.45 - **40.77** |
| 0.004 | 40.28 - **41.11** | 40.45 - **40.93** | 40.38 - **40.86** | 40.39 - **40.79** |
| 0.009 | 40.41 - **41.17** | 40.55 - 40.50 | 40.35 - **40.75** | 40.27 - **40.85** |
| 0.04 | 40.39 - **41.20** | 40.40 - **40.89** | 40.35 - **40.93** | 40.44 - **40.78** |
| 0.09 | 40.35 - **41.17** | 40.32 - **40.95** | 40.40 - **40.90** | 40.46 - **40.84** |
| 0.4 | 40.42 - **41.22** | 40.32 - **40.98** | 40.43 - **40.90** | 40.29 - **40.82** |
| 0.9 | 40.39 - **41.15** | 40.34 - **41.03** | 40.21 - **40.86** | 40.10 - **40.60** |

Table 7: Average MaxIS size in the format "without–with" the clique term, across different values of step size $\alpha$ and edge penalty parameter $\gamma$. Bold results correspond to the cases where pCQO-MIS obtained better results than the best of pQO (underlined).

The following are the key observations from Table 7 for which the bold results correspond to cases where using the clique term resulted in a MaxIS size higher than the best of the "without" case (underlined):

1. As observed, the difference between best pCQO (with) and the best pQO (without) is nearly 4 nodes which is similar to what we report in Table 1, nearly 4.1 nodes on average.

2. When $\gamma' = 7$, our approach returns better results across learning rates and $\gamma$'s compared to $\gamma' = 0$. Additionally, $\gamma' = 0$ is not competitive compared with any of the baseline solvers we tested in this paper, as it achieves at most 40.55 (the underlined result in the table). Only when the clique term is introduced does our method become competitive with other solvers.

3. Out of all combinations above, there are only two cases where $\gamma' = 0$ is slightly better.

In addition to average MaxIS size, we evaluated how many optimizer steps were required to obtain the first MaxIS solution for each set of parameters. The results are reported as the average time to first solution over the ER dataset in Table 8. In all cases, $\gamma' = 7$ finds a viable solution first. We conjecture that, due to the presence of the third clique term, a "smoother" optimization landscape is created for each of the evaluated hyperparameter sets.

We note that the above results indicate that there might exist a set of hyper-parameters with no MC term that result in a better MaxIS when compared to using the MC term. However, from our experiments, we only obtain the competitive results with baselines when the MC term is included.

### E.6. Comparison with a Clique Heuristic Solver

In this subsection, we include comparison results of 31 graphs (from DIMACS dataset) with an efficient clique heuristic solver called the Minimal Independent Set based Approach (MISB) (Singh & Govinda, 2014), which demonstrated competitive performance on these graphs with $n \leq 500$.

Table 9 shows the results of 5 graphs as examples. The complete table can be found online[5]. The results of MISB is sourced from Table 1 of (Singh & Govinda, 2014). It can be seen that our algorithm consistently outperforms MISB and achieves optimal or near-optimal solution. Here, $\rho$ is the graph density.

---

[5] https://github.com/ledenmat/pCQO-mis-benchmark/blob/main/Comparison_with_MSIB_MC_Solver.pdf

| Step size $\alpha$ | $\gamma = 350.0$ | $\gamma = 450.0$ | $\gamma = 525.0$ | $\gamma = 600.0$ |
|---|---|---|---|---|
| 4e-06 | N/A - 425.00 | N/A - 442.00 | N/A - 436.45 | N/A - **434.79** |
| 9e-06 | N/A - 261.44 | N/A - 244.73 | N/A - 234.52 | N/A - **222.99** |
| 4e-05 | N/A - **73.29** | N/A - **71.94** | N/A - **71.35** | N/A - **70.45** |
| 9e-05 | N/A - **46.26** | N/A - **47.54** | N/A - **48.34** | N/A - **48.91** |
| 0.0004 | 206.95 - **29.82** | 208.50 - **31.48** | 209.69 - **32.80** | 210.86 - **33.79** |
| 0.0009 | 134.53 - **26.63** | 136.81 - **28.56** | 138.15 - **29.88** | 139.29 - **30.82** |
| 0.004 | 89.53 - **24.00** | 91.95 - **26.09** | 93.42 - **27.48** | 94.66 - **28.65** |
| 0.009 | 81.19 - **23.41** | 83.63 - 25.62 | 85.07 - **27.04** | 86.31 - **28.38** |
| 0.04 | 74.05 - **23.31** | 76.39 - **25.46** | 77.86 - **26.92** | 79.20 - **28.11** |
| 0.09 | 72.23 - **23.38** | 74.66 - **25.49** | 76.05 - **26.98** | 77.31 - **28.14** |
| 0.4 | 70.51 - **23.33** | 72.91 - **25.43** | 74.32 - **26.90** | 75.55 - **28.16** |
| 0.9 | 69.92 - **23.34** | 72.27 - **25.42** | 73.71 - **26.92** | 74.99 - **28.15** |

Table 8: Average number of steps to converge in the format "without–with" the clique term, for various step sizes $\alpha$ and edge penalties $\gamma$. Bold results follow Table 7.

| Graph Name | $n$ | $m$ | Density | **Optimal** | **Ours** | **MISB MC Solver** (Singh & Govinda, 2014) |
|---|---|---|---|---|---|---|
| cc-fat500-2 | 500 | 115611 | 0.92 | 26 | **26** | 26 |
| p_hat300-2 | 300 | 22922 | 0.51 | 25 | **25** | 24 |
| sanr200_0.7 | 200 | 6032 | 0.30 | 18 | **18** | 16 |
| brock400_1 | 400 | 20077 | 0.25 | 27 | **25** | 23 |
| gen200_p0.9_55 | 200 | 1990 | 0.10 | 55 | **55** | 49 |

Table 9: Comparison between pCQO-MIS and MISB clique solver.

We note that in the recent survey paper about clique solvers (Marino et al., 2024), the authors recognized ReduMIS (Lamm et al., 2016) (the main heuristic we compare with in our paper) as "extremely effective" for solving the clique problem (see Section 3.3.1) when compared to other methods.

### E.7. Comparison with the Relu-based Dataless Solver

Here, we compare pCQO-MIS with the dataless Neural Network (dNN) MIS solver in (Alkhouri et al., 2022). In this experiment, we use 10 GNM graphs with $(n, m) = (100, 500)$ and report the average MIS size and average run-time (in seconds) for solving one initialization. The results are given in Table 10. As observed, pCQO-MIS outperforms the dNN-MIS method in (Alkhouri et al., 2022) in terms of both the run-time and MIS size.

| Method | Average MIS Size | Average Run-Time (seconds) |
|---|---|---|
| dNN-MIS (Alkhouri et al., 2022) | 27.4 | 24 |
| **pCQO-MIS** (Ours) | 29.9 | 0.7 |

Table 10: Evaluation of pCQO-MIS vs. the MIS dNN solver in (Alkhouri et al., 2022) in terms of MIS size and run-time (seconds) over 10 GNM graphs with $(n, m) = (100, 500)$.

### E.8. Comparison with Leading data-centric Solver with Different Densities

In this subsection, we compare our approach with the leading data-driven baseline, DIFUSCO. DIFUSCO uses a pre-trained diffusion model trained on ER700-800 graphs (with $p = 0.15$) labeled using ReduMIS.

Here, we compare pCQO-MIS to DIFUSCO using graphs (with $n = 700$) with varying edge creation probabilities, $p$. The results, presented in Table 11, are averaged over 32 graphs for each $p$, with DIFUSCO utilizing 4-sample decoding. For pCQO-MIS, hyperparameters remain fixed across all values of $p$.

As observed, our method consistently outperforms DIFUSCO in both average MIS size and run-time. Notably, our run-time remains constant as the number of edges increases, supporting our claim that the run-time scales only with the number of nodes in the graph. DIFUSCO reports relatively smaller MIS sizes, particularly for $p = 0.05$ and $p = 0.2$, which are slightly different from the training graphs. This underscores the generalization limitations of a leading learning-based method.

| Probability of Edge Creation $p$ | DIFUSCO (Sun & Yang, 2023) | | pCQO-MIS (Ours) | |
|---|---|---|---|---|
| | Avg. MIS Size ($\uparrow$) | Run-time ($\downarrow$) | Avg. MIS Size ($\uparrow$) | Run-time ($\downarrow$) |
| 0.05 | 88.25 | 4.62 | 97.34 | 4.73 |
| 0.10 | 58 | 8.63 | 59.25 | 4.71 |
| 0.15 (Training setting of DIFUSCO) | 40.81 | 12.98 | 43.2 | 4.67 |
| 0.2 | 29.22 | 17.66 | 33.78 | 4.45 |

Table 11: Evaluation of pCQO-MIS vs. the ER700-trained DIFUSCO (with $p = 0.15$) in (Sun & Yang, 2023) in terms of average MIS size and sequential run-time (minutes) over 32 ER graphs for each $p$.

### E.9. pCQO-MIS Hyper-Parameters

In this subsection, we outline the pCQO-MIS parameters (i.e., $\gamma$, $\gamma'$, $\alpha$, $\beta$, $T$, and $\eta$) used in the paper, along with examples from the tuning procedure conducted to select these parameters.

Table 12 provides the specific parameter values used for Table 1 and Figure 2 in Section 4. These hyper-parameters are selected based on a grid search as those provided in Table 13 and Table 14 for the ER dataset. The captions of these tables provide the parameters we fix and the parameters we vary, and in both cases, we report the average MIS size of 6 ER graphs. Other than the first three columns of the last row of Table 13, the reported average MIS size (in both tables) vary between 37.67 and 41.83. This indicates that pCQO-MIS results do not vary significantly with the choice of these parameters in term of finding feasible solutions.

| Graph Dataset | Edges-penalty $\gamma$ | MC parameter $\gamma'$ | Step size $\alpha$ | Momentum $\beta$ | Steps $T$ | Exploration parameter $\eta$ |
|---|---|---|---|---|---|---|
| SATLIB | 900 | 1 | $3e-4$ | 0.875 | 30 | 2.25 |
| ER | 350 | 7 | $9e-6$ | 0.9 | 450 | 2.25 |
| GNM with $n \in \{50, 500, 1000\}$ | 100 | 5 | $1e-2$ | 0.55 | 200 | 1 |
| GNM with $n \in \{1500, 2000\}$ | 100 | 10 | $1e-2$ | 0.55 | 200 | 1 |

Table 12: Hyper-parameters for pCQO-MIS used in Section 4. This selection is made based on ablation studies such as those in Table 13 and Table 14.

| Step Size $\alpha$ | $\beta = 0.1$ | $\beta = 0.5$ | $\beta = 0.7$ | $\beta = 0.9$ |
|---|---|---|---|---|
| $1e-2$ | 41.83 | 38.83 | 38.17 | 39.83 |
| $5e-3$ | 42.00 | 38.83 | 37.50 | 40.17 |
| $1e-3$ | 41.17 | 38.67 | 38.17 | 39.67 |
| $5e-4$ | 40.83 | 39.00 | 38.67 | 40.00 |
| $1e-4$ | 37.67 | 41.17 | 39.67 | 41.00 |
| $5e-5$ | 38.33 | 41.50 | 40.50 | 40.00 |
| $1e-5$ | 5.67 | 35.33 | 17.83 | 40.67 |

Table 13: Average MIS size of 6 ER graphs for different values of $\alpha$ and $\beta$. Here, $\gamma = 300$, $\gamma' = 1$, and $T = 300$. The initialization of $\mathbf{x}[0]$ is $\mathbf{h}$ in Eq. (13).

| Edges Penalty $\gamma$ | MC Term $\gamma'$ | pCQO-MIS (MaxIS Size) |
|---|---|---|
| 300 | 1 | 40.67 |
| 300 | 5 | 40.16 |
| 500 | 1 | 39.83 |
| 500 | 5 | 40.33 |
| 775 | 1 | 39.33 |
| 775 | 5 | 39.67 |

Table 14: Average MaxIS size of 6 ER graphs using different values of $\gamma$ and $\gamma'$. Here, $\alpha = 1e-5$, $\beta = 0.9$, and $T = 300$. The initialization of $\mathbf{x}[0]$ is $\mathbf{h}$ in Eq. (13).

#### E.9.1. A BASIC WARM START PROCEDURE FOR HYPER-PARAMETER TUNING

Here, we describe the procedure the grid search we used for our hyper-parameter tuning. We first perform a grid search over the following parameters using $T = 750$. We use $\alpha \in \{0.5, 0.05, 0.005, 0.0005, 0.0001, 0.00001, 0.000001, 0.0000001\}$, $\beta \in \{0.99, 0.9, 0.75\}$, $\gamma \in \{250, 500, 1000, 2000, 5000\}$, and $\gamma' \in \{1, 3, 5\}$. Based on the results of these combinations,

we use a second loop that takes all the best choices from the grid search and reduces $T$ until it impacts solution size. This procedure takes approximately 30 seconds to run on one ER graph.

To run our method on new graphs, we recommend using the parameters tuning grid search.

### E.10. Results of Table 1 based on the Number of Batches

In this subsection, we provide the main pCQO-MIS results based on the number of batches. Table 15 (resp. Table 16) presents the results for the SATLIB (resp. ER) dataset. The results of Table 1 are obtained from these tables.

| Batches Solved | pCQO-MIS (MIS Size) | pCQO-MIS (Run time) |
|---|---|---|
| 1 | 408.286 | 0.408 |
| 10 | 417.228 | 2.454 |
| 20 | 420.276 | 4.726 |
| 30 | 421.610 | 6.996 |
| 40 | 422.456 | 9.265 |
| 50 | 422.988 | 11.533 |
| 60 | 423.400 | 13.799 |
| 70 | 423.706 | 16.065 |
| 80 | 423.930 | 18.329 |
| 90 | 424.096 | 20.593 |
| 100 | 424.278 | 22.856 |
| 110 | 424.406 | 25.119 |
| 120 | 424.508 | 27.380 |
| 130 | 424.606 | 29.641 |
| 140 | 424.686 | 31.901 |
| 150 | 424.736 | 34.161 |
| 160 | 424.798 | 36.419 |
| 170 | 424.856 | 38.678 |
| 180 | 424.906 | 40.935 |
| 190 | 424.950 | 43.191 |
| 200 | 425.006 | 45.448 |
| 210 | 425.032 | 47.704 |
| 220 | 425.064 | 49.959 |
| 230 | 425.098 | 52.214 |
| 240 | 425.126 | 54.468 |
| 250 | 425.148 | 56.722 |

Table 15: pCQO-MIS SATLIB results (average MIS size and total run time in minutes) including the number of batches used (column 1).

| Batches Solved | pCQO-MIS (MIS Size) | pCQO-MIS (Run time) |
|---|---|---|
| 1 | 39.344 | 0.153 |
| 10 | 43.086 | 1.159 |
| 20 | 43.836 | 2.266 |
| 30 | 44.117 | 3.367 |
| 40 | 44.367 | 4.466 |
| 50 | 44.500 | 5.563 |
| 60 | 44.578 | 6.659 |
| 70 | 44.656 | 7.753 |
| 80 | 44.695 | 8.848 |
| 90 | 44.750 | 9.942 |
| 100 | 44.789 | 11.035 |
| 110 | 44.828 | 12.129 |
| 120 | 44.836 | 13.222 |
| 130 | 44.859 | 14.315 |
| 140 | 44.875 | 15.409 |
| 150 | 44.898 | 16.502 |
| 160 | 44.914 | 17.595 |
| 170 | 44.938 | 18.688 |
| 180 | 44.961 | 19.781 |
| 190 | 44.969 | 20.875 |
| 200 | 44.977 | 21.968 |
| 210 | 44.984 | 23.062 |
| 220 | 45.000 | 24.155 |
| 230 | 45.016 | 25.249 |
| 240 | 45.023 | 26.342 |
| 250 | 45.023 | 27.435 |
| 260 | 45.031 | 28.528 |
| 270 | 45.039 | 29.622 |
| 280 | 45.039 | 30.715 |
| 290 | 45.047 | 31.809 |
| 300 | 45.055 | 32.902 |
| 310 | 45.062 | 33.996 |
| 320 | 45.070 | 35.089 |
| 330 | 45.078 | 36.182 |
| 340 | 45.078 | 37.276 |
| 350 | 45.078 | 38.369 |
| 360 | 45.078 | 39.462 |
| 370 | 45.078 | 40.555 |
| 380 | 45.078 | 41.648 |
| 390 | 45.078 | 42.741 |
| 400 | 45.094 | 43.834 |
| 410 | 45.094 | 44.928 |
| 420 | 45.094 | 46.021 |
| 430 | 45.094 | 47.115 |
| 440 | 45.094 | 48.208 |
| 450 | 45.102 | 49.301 |
| 460 | 45.102 | 50.393 |
| 470 | 45.102 | 51.486 |
| 480 | 45.102 | 52.579 |
| 490 | 45.102 | 53.672 |
| 500 | 45.109 | 54.766 |

Table 16: pCQO-MIS ER results (average MIS size and total run time in minutes) including the number of batches used (column 1).

