# OpenReview forum: "Differentiable Quadratic Optimization For the Maximum Independent Set Problem"
_ICML.cc/2025/Conference — ICML 2025 poster_

### Official Review · Reviewer_sDzA · 2025-03-10

**Overall Recommendation:** 3

**Summary:**

This paper proposes a novel solution for the Maximum Independent Set Problem (MIS), a well-known NP-hard problem. The proposed method extends the quadratic formulation by [Pardalos & Rodgers, 1992] by introducing a max clique term into the non-convex objective function. The optimization is performed using projected momentum-based gradient descent, and a rounding step is applied to obtain a candidate solution for MIS. The method is designed to leverage GPU-based parallel execution.

The authors conduct experiments on Erdős-Rényi (ER) graph datasets and SATLIB graph datasets, demonstrating that the proposed approach outperforms existing learning-based methods. Additionally, on high-density graphs, the method achieves superior performance compared to exact solvers such as Gurobi and CP-SAT, as well as heuristic solvers like ReduMIS.

**Claims And Evidence:**

The authors' claims are clear throughout the paper and are supported both theoretically and experimentally.

**Essential References Not Discussed:**

All important citations have been properly included without any issues.

**Experimental Designs Or Analyses:**

I have reviewed the experimental setup and did not identify any major issues.

**Methods And Evaluation Criteria:**

The proposed method and evaluation criteria are meaningful. While the datasets used in the main text feel somewhat insufficient, the additional experiments in the appendix make the overall evaluation satisfactory.

**Other Comments Or Suggestions:**

* A.3. Proof of Lemma 9 in appendix should be A.3. Proof of Lemma 8.

**Other Strengths And Weaknesses:**

Overall, this paper is well-structured and highly readable. The approach is straightforward and convincing in relation to its objective, and the claims are well-supported both theoretically and experimentally.

The weaknesses are as follows.

* I did not fully understand why the maximum clique term in Equation (3) leads to such significant improvements. The authors provide an explanation on the right side of line 124, but it does not seem entirely convincing in justifying the substantial improvements observed in Table 1. Further investigation of this aspect, possibly using additional experimental results, could strengthen the credibility of the approach.

* In Appendix B, it is stated that non-extremal stationary points $x'$ are rarely contained within the box constraint, and even if they are, they appear at most at a single point. In this case, the ability of MGD to escape saddle points, as discussed in Remark 13, might not be very useful. If this is the case, the motivation for using MGD, as described in Remark 13, seems less convincing.

* The experimental results suggest that the proposed method performs particularly well on high-density graphs (i.e., graphs with a large number of edges) compared to existing exact solvers and heuristic approaches. However, how common is it to need to find independent sets in such high-density graphs? In practical applications, aren't graphs often less dense? It would be helpful if the authors could clarify the intended application areas of their approach. That being said, I do not mean to suggest that developing strong methods for high-density graphs is unimportant in this research field.

* The authors have conducted a sensitivity analysis of the parameters in the proposed method, which is commendable. However, in Section D.5, they state:
"Other than the first three columns of the last row of Table 7, the reported average MIS size (in both tables) varies between 37.67 and 41.83. This indicates that pCQO-MIS is not very sensitive to the choice of these parameters."
Contrary to this claim, I got the impression that the output is relatively sensitive to parameter choices. This is because, in the Maximum Independent Set problem, once a provisional solution is found, increasing the solution size by even one can be quite difficult. As a result, the variation between 37 and 42 could indicate a significant difference in solution quality.
Furthermore, from Table 6, it appears that the parameters need to be adjusted for each problem instance. If these parameters must be determined through an ablation study, wouldn’t this require additional computational cost for the proposed method?

**Questions For Authors:**

Please see Other Strengths And Weaknesses.

**Relation To Broader Scientific Literature:**

This paper contributes to an important research topic in the field of combinatorial optimization, namely the Maximum Independent Set problem, by developing a novel differentiable solver. The proposed method is simple, achieves favorable experimental results, and is considered to make a sufficient contribution.

**Theoretical Claims:**

I conducted a brief verification of the theoretical claims and did not find any particular issues.

---

> ### Author Rebuttal · Authors · 2025-04-01
>
> We would like to thank the reviewer for their comments. Please refer to (https://anonymous.4open.science/r/pCQO-mis-benchmark-81AF/Tables_rebuttal.md) for Tables A, B, & C.
>
> We are glad that the reviewer finds the paper to be well-structured & highly readable & that our claims are clear & supported both theoretically & experimentally.
>
> ### (W1) **Additional results for the clique term**.
>
> The clique term is introduced to (i) encourage the optimizer to select two nodes connected by an edge in the complement graph, leveraging the MIS & MC duality, & (ii) to discourage sparsity in the solution given the $\ell_1$ norm. We note that the computational cost for including the third term is the same as not including it (Remark 3).
>
> We have empirically observed that the third term improves the optimization process by enhancing stability, preventing overshooting, & leading to better minima.
>
> Table A compares the cases *without* (pQO) & *with* (pCQO) the clique term ($\gamma' = 0$ vs. $\gamma' = 7$) over the ER dataset used in Table 1. The results are presented as the average MIS size in the format "without–with" & are obtained across different values of $\alpha$ (learning rate) & $\gamma$ (edge penalty parameter). The results are reported after optimizing 50 batches of initializations for each set of hyper-parameters. We note that the range of $\gamma$ is selected based on the criteria in Theorem 9.
>
> Key observations from Table A (bold results correspond to cases where pCQO resulted in a MaxIS size higher than the best of the pQO case (underlined)) are:
>
> 1. The difference between pCQO & pQO is nearly 4 nodes which is similar to what we report in Table 1.
>
> 2. pCQO approach returns better results across most $\alpha$'s' & $\gamma$'s' compared to pQO. Additionally, $\gamma' = 0$ is not competitive compared with most other baselines, as it achieves at most 40.55 (the underlined result in the table). Only when the clique term is introduced does our method become competitive with other solvers.
>
> 3. Out of all combinations above, there are only two cases where pQO is slightly better.
>
> Additionally, we evaluated how many optimizer steps were required to obtain the first MaxIS solution for pCQO & pQO. See Table B. In all cases, pCQO finds a viable solution first. We conjecture that, due to the presence of the third clique term, a "smoother" optimization landscape is created for each of the evaluated hyperparameter sets. We will include the above experiments in the revised version of the paper.
>
> ### (W2) **The motivation of MGD.**
>
> Extremal stationary points depend on graph connectivity, as noted in Appendix C. However, our use of MGD is not solely to escape these points but also due to the empirical observation that, from the same initial point, MGD converges to minimizers with larger MaxIS values while avoiding the overshooting seen in vanilla GD. Additionally, momentum generally accelerates GD convergence. We will revise the remark to clarify this point.
>
> In Table C, we use 5 ER graphs with $n=100$ & $p\in \{{0.3,0.6\}}$ (probability of edge creation) & run GD vs. MGD, **using the exact same $\gamma, \gamma', \alpha,$ & the initializations.** As observed, on average, MGD converges to larger MIS. Furthermore, MGD avoids the all 0's which is the case of overshooting in GD.
>
>
> ### (W3) **Large high-density graphs.**
>
> We appreciate the reviewer’s comment. The main advantage of our solver is most evident for dense, large graphs, where (i) ILPs are impractical due to extensive run-time, & (ii) ReduMIS fails to significantly reduce graph size. Our motivation is to address these challenges rather than focus on a specific application.
>
> We will clarify this in the revised paper & leave application-specific explorations for future work. MIS has broad applications in scheduling, genome sequencing, & fault detection, often involving large, dense graphs. For instance, [A] applies conflict graph constructions in genome sequencing.
>
> Even sparse graphs can become dense in dynamic settings.
>
> [A] On the Maximal Cliques in C-Max-Tolerance Graphs & Their Application in Clustering Molecular Sequences
>
> ### (W4) **Sensitivity to hyper-parameters.**
>
> Thank you for your comment. When we stated that our method is not very sensitive to the selection of these hyperparameters, we meant that a range of values can yield feasible solutions. In other words, obtaining results with our method does not heavily depend on precise hyper-parameter tuning, as long as they satisfy the condition in Theorem 9. We will refine this claim to better align with our intended meaning.
>
> That said, we agree with the reviewer that achieving the best results require hyper-parameter tuning. We note that we do hyper-parameter tuning on one instance & apply it on the other graphs. This means that the additional running time of each graph increases by the tuning time divided by the total number of graphs in the dataset. We will include this discussion in the revised paper.

---

### Official Review · Reviewer_keSC · 2025-03-12

**Overall Recommendation:** 4

**Summary:**

The paper introduces pCQO-MIS, a new quadratic optimization approach for solving the Maximum Independent Set (MIS) problem. By incorporating a maximum clique (MC) term, the method improves convergence and exploration, using parallelized momentum-based gradient descent to efficiently find maximal independent sets. The authors provide theoretical results ensuring that local minimizers correspond to maximal independent sets and present an efficient procedure for MIS checking. Experimental results show that pCQO-MIS outperforms existing exact, heuristic, and data-driven methods, achieving larger MIS sizes and faster runtimes, particularly for denser graphs, without the need for training data.

**Claims And Evidence:**

The proposed method is well motivated from the duality of MIS and clique problem. The authors prove the correctness of the method ensuring that local minimizers correspond to maximal independent sets.

**Essential References Not Discussed:**

No

**Experimental Designs Or Analyses:**

Yes, The authors evaluate the performer of the method with comparison to several baselines on graphs with different density. Comparison on both quality and run time are provided. Just wonder how the proposed algorithm compares to clique-based method?

**Methods And Evaluation Criteria:**

The proposed algorithm together with the efficient MIS checker makes the proposed method both efficient and effective. The authors evaluate the performer of the method with comparison to several baselines on graphs with different density. Comparison on both quality and run time are provided.

**Other Comments Or Suggestions:**

N/A

**Other Strengths And Weaknesses:**

N/A

**Questions For Authors:**

N/A

**Relation To Broader Scientific Literature:**

The MIS is a NP-problem with wide applicability.

**Theoretical Claims:**

No

---

> ### Author Rebuttal · Authors · 2025-04-01
>
> We would like to thank the reviewer for their comments.
>
> We are glad that the reviewer finds our method well-motivated. We also appreciate the reviewer acknowledging that we prove the correctness of the method, ensuring that local minimizers correspond to maximal independent sets.
>
> Please see our response below regarding the comparison with a clique solver.
>
> ### (C) **Comparison with a clique-based method.**
>
> We thank the reviewer for their comment.
>
> In the recent survey paper about the MC solvers [A], the authors recognized ReduMIS (the heuristic we compare against in our submission) as **"extremely effective"** for solving the clique problem (see Section 3.3.1 of [A]) when compared to other methods.
>
> To fully address the reviewer's comment, we have included comparison results of 31 graphs (from DIMACS) with an MC heuristic solver called MISB [B], which demonstrated competitive performance on the selected graphs with $n\leq 500$.
>
> Below we show the results of 5 graphs as examples. The complete results can be found in (https://anonymous.4open.science/r/pCQO-mis-benchmark-81AF/Comparison_with_MSIB_MC_Solver.pdf). The results of MISB is sourced from Table 1 of [B]. It can be seen that our algorithm *consistently outperforms* MISB and achieves optimal or near-optimal solution in these cases. Here, $\rho$ is the graph density.
>
> | Graph Name |  $n$  | $m$    | $\rho$ | Optimal | Ours  | MISB Clique Solver [B] |
> |-------------|----------|--------|--------------|-|-|-|
> | cc-fat500-2 | 500 |	115611 | 0.92 | 26 | **26** |  26 |
> | p_hat300-2 | 300 |22922 | 0.51 | 25 | **25** |  24 |
> | sanr200_0.7 | 200 |6032 | 0.3 | 18 | **18** |  16 |
> | brock400_1 | 400 |20077| 0.25 | 27 | **25** |  23 |
> | gen200_p0.9_55 | 200 |1990| 0.1 | 55 | **55** |  49 |
>
> We plan to include additional comparisons in the camera-ready version if our paper is accepted.
>
> [A] A Short Review On Novel Approaches For Maximum Clique Problem: Form Classical Algorithms to Graph Neural Networks and Quantum Algorithms. March 2024. (https://arxiv.org/pdf/2403.09742v1)
> [B] A simple and efficient heuristic algorithm for maximum clique problem. ISCO 2014.

---

### Official Review · Reviewer_HPTx · 2025-03-13

**Overall Recommendation:** 3

**Summary:**

In solving the Maximum Independent Set (MIS) problem, the authors propose its continuous relaxation as an optimization problem of a quadratic differentiable function, which can be solved by first-order gradient-based method starting from multiple parallel initial points. The proposed method named parallelized Clique-Informed Quadratic Optimization of MIS (pCQO-MIS) is compared with other MIS solvers.

## Update after rebuttal

My concerns and questions about the paper have been clarified, and I intend to maintain my original recommendation score.

**Claims And Evidence:**

The continuous relaxation of the MIS problem seems to be valid.

**Essential References Not Discussed:**

No special reference I would like to add.

**Experimental Designs Or Analyses:**

The experiment presented is comparison of benchmarks between pQQ-MIS and other existing MIS solvers. All procedure seems valid.

**Methods And Evaluation Criteria:**

The experiment presented is comparison of benchmarks between pQQ-MIS and other existing MIS solvers. All procedure seems valid.

**Other Comments Or Suggestions:**

Regarding to (4), $x^TJ_nx = \Vert x \vert_1^2$ does not hold if any of the element of $x$ is negative; but this does not seem to be relevant with the main arguments.

**Other Strengths And Weaknesses:**

As the optimization method is based on the first-order gradient method, the proposed algorithm has advantage on the scalability, showing less run-time on denser and bigger graph. However, the heuristic or the exact method seems to be more beneficial in graphs with less number of edges and vertices.

Also, good selection of hyperparameter $\gamma$ is essential in finding the correct solution, thus solving the problem repeatedly with different hyperparameters is unavoidable in practice.

**Questions For Authors:**

1. Can you give more detailed explanation about why (9) indicates MaxIS?

2. In Table 1, the run-time of the exact method differ too much between the two datasets. I am suspecting that this phenomenon happens because in the number of constraints $z_v + z_u \leq 1, \forall (v,u) \in E$ in the integer linear program becomes too many in the dense ER dataset. Is there a way to work around this constraint rather than listing all constraint for all edges?

**Relation To Broader Scientific Literature:**

The proposed optimization strategy, MGD, is a first-order gradient-based optimization which has a benefit in scalability. In this sense, we can think about an optimization strategy that utilizes multiple parallel computational nodes when the given $\mathbf{A}_{G}$ is too big to fit in a single machine.

**Theoretical Claims:**

The main theoretical claims seem to be correct.

---

> ### Author Rebuttal · Authors · 2025-04-01
>
> We would like to thank the reviewer for their comments. Below is a point-by-point response.
>
> ### (C1) **As the optimization method is based on the first-order gradient method, the proposed algorithm has advantage on the scalability, showing less run-time on denser and bigger graph. However, the heuristic or the exact method seems to be more beneficial in graphs with less number of edges and vertices**
>
> Utilizing a first-order gradient method indeed allows our method to scale to larger graphs, which is particularly evident in relatively denser graphs, as shown in the results of Figure 2.
>
> However, as the reviewer correctly points out, for sparse graphs, methods like ReduMIS and ILP solvers tend to yield better solutions in terms of both MIS size and run-time. In ILP solvers, the number of constraints exactly equals the number of edges in the graph. Since sparser graphs have fewer edges, these solvers are able to find solutions more quickly. Similarly, ReduMIS benefits from a greater effectiveness of MIS-specific graph reductions it employs when operating on sparser graphs.
>
> ### (C2) **Good selection of hyperparameter $\gamma$ is essential in finding the correct solution, thus solving the problem repeatedly with different hyperparameters is unavoidable in practice.**
>
> We agree with the reviewer that selecting $\gamma$ according to Theorem 9 is important for guaranteeing that all local minimizers are feasible MaxISs.
>
> To obtain the best solutions in our method, hyper-parameters fine-tuning is needed in some cases. An example of such a fine-tuning procedure is discussed in Tables 7 and 8 of Appendix D.5.
>
> We say "in some cases" because in our evaluation on the DIMACS graphs in Table 3, we applied a set of hyperparameters optimized on one graph of the dataset, across the entire dataset and obtained the optimal solution in 49 out of 61 graphs. Note that these graphs are very diverse and vary not only in terms of graph order but also density (see columns 2 to 4 of Table 3).
>
> Hyper-parameter tuning is not a major bottleneck as there have been several studies for automatic and efficient hyper-parameter tuning (such as bilevel optimization [A]) and our method can be integrated with them.
>
> [A] Franceschi et al, "Bilevel Programming for Hyperparameter Optimization and Meta-Learning" ICML2018
>
> ### (C3) **Regarding to (4), $x^T ee^T x = \||x\||^2_1$ does not hold if any of the element of is negative; but this does not seem to be relevant with the main arguments.**
>
> The reviewer is indeed correct. Thanks for pointing this out. Given that $x\in [0,1]^n$, there won't be negative elements. To be more rigorous, in the revised paper, we will add the following sentence before presenting Eq. (4). "For $x\in [0,1]^n$, we can write $x^T ee^T x = \||x\||^2_1$ and therefore (introduce equation 4)".
>
> ### (Q1) **Can you give more detailed explanation about why (9) indicates MaxIS?**
>
> Thank you for your question. The condition in Eq. (9) checks whether some $z\in \{0,1\}^n$ is a **fixed point or not** given a projected gradient descent step. This is based on our characterizations of the proposed objective, which indicates that all binary fixed points are local minimizers and therefore MaxISs.
>
> We reach this conclusion by showing that:
>
> 1. All local minimizers are binary (Lemma 8), and
> 2. All local minimizers are MaxISs if $\gamma > 1 + \gamma' \Delta(G')$ (Theorem 9)
>
> Another reason that Eq. (9) can be used for checking MaxIS is due to Theorem 12: When $x'$ such that $\nabla_xf(x')=0$ exists, then it is also a fixed point. However, in the proof of Theorem 12, we show that $x'$ can not be binary. Therefore, in Eq. (9), we only check $z\in \{0,1\}^n$ which is the binarized version of $x$. This means that if the optimizer reaches a binary fixed point, then it must be MaxIS.
>
> We hope that we have answered the reviewer's question. Please let us know if it remains unclear.
>
> ### (Q2) **In Table 1, the run-time of the exact method differ too much between the two datasets. I am suspecting that this phenomenon happens because in the number of constraints in the integer linear program becomes too many in the dense ER dataset. Is there a way to work around this constraint rather than listing all constraint for all edges?**
>
> The reviewer is correct. The main reason for exact methods taking longer times on denser graphs is that, in the MIS problem, the number of constraints in the ILP exactly equals the number of edges in $G$. Cutting plane methods [B] may be used to reduce the number of constraints in the ILP, but such a method for the exact ILP methods is beyond the scope of our paper, as we focus on differentiable approaches.
>
> [B] G. Nemhauser and L. Wolsey, "Integer and Combinatorial Optimization," Wiley 1998.

---

> > ### Comment · Reviewer_HPTx · 2025-04-07
> >
> > Thank you for your thorough and detailed response. My concerns and questions about the paper have been clarified, and I intend to maintain my original recommendation score.

---

### Decision · Program_Chairs · 2025-05-01

**Decision:**

Accept (poster)

**Comment:**

The authors develop a novel gradient-based optimization problem for the max independent set problem, which incorporates an additional term corresponding to the complementary graph’s max clique problem, scaled by a hyperparameter gamma. This results in an optimization problem that is non-convex but amenable to modern gradient optimization techniques, and the authors parallelize the optimization over initializations and settings of gamma. Experimentally, they show that their method is competitive with various state of the art solvers, but is more computationally scalable in settings with more edges.

Overall, reviewers were generally positive, viewing the work as an interesting practical development on an important problem, which provides a benefit for a certain type of scalability.  Authors agreed to include new experiments comparing at least one max clique solver, as well.